# NEURAL PROCESSING OF TRI-PLANE HYBRID NEURAL FIELDS

**Adriano Cardace**[1]    **Pierluigi Zama Ramirez**[1]    **Francesco Ballerini**[1]    **Allan Zhou**[2]
**Samuele Salti**[1]    **Luigi Di Stefano**[1]
[1] University of Bologna    [2] Stanford University
`adriano.cardace2@unibo.it`

## ABSTRACT

Driven by the appealing properties of neural fields for storing and communicating 3D data, the problem of directly processing them to address tasks such as classification and part segmentation has emerged and has been investigated in recent works. Early approaches employ neural fields parameterized by shared networks trained on the whole dataset, achieving good task performance but sacrificing reconstruction quality. To improve the latter, later methods focus on individual neural fields parameterized as large Multi-Layer Perceptrons (MLPs), which are, however, challenging to process due to the high dimensionality of the weight space, intrinsic weight space symmetries, and sensitivity to random initialization. Hence, results turn out significantly inferior to those achieved by processing explicit representations, e.g., point clouds or meshes. In the meantime, hybrid representations, in particular based on tri-planes, have emerged as a more effective and efficient alternative to realize neural fields, but their direct processing has not been investigated yet. In this paper, we show that the tri-plane discrete data structure encodes rich information, which can be effectively processed by standard deep-learning machinery. We define an extensive benchmark covering a diverse set of fields such as occupancy, signed/unsigned distance, and, for the first time, radiance fields. While processing a field with the same reconstruction quality, we achieve task performance far superior to frameworks that process large MLPs and, for the first time, almost on par with architectures handling explicit representations.

## 1 INTRODUCTION

**A world of neural fields.** Neural fields (Xie et al., 2021) are functions defined at all spatial coordinates, parameterized by a neural network such as a Multi-Layer Perceptron (MLP). They have been used to represent different kinds of data, like image intensities, scene radiances, 3D shapes, etc. In the context of 3D world representation, various types of neural fields have been explored, such as the signed/unsigned distance field (*SDF*/*UDF*) (Park et al., 2019; Chibane et al., 2020; Gropp et al., 2020; Takikawa et al., 2021), the occupancy field (*OF*) (Mescheder et al., 2019; Peng et al., 2020), and the radiance field (*RF*) (Mildenhall et al., 2020b). Their main advantage is the ability to obtain a continuous representation of the world, thereby providing information at every point in space, unlike discrete counterparts like voxels, meshes, or point clouds. Moreover, neural fields allow for encoding a 3D geometry at arbitrary resolution while using a finite number of parameters, i.e., the weights of the MLP. Thus, the memory cost of the representation and its spatial resolution are decoupled.

Recently, hybrid neural fields (Xie et al., 2021), which combine continuous neural elements (i.e., MLPs) with discrete spatial structures (e.g., voxel grids (Peng et al., 2020), point clouds (Tretschk et al., 2020), etc.) that encode local information, are gaining popularity due to faster inference (Reiser et al., 2021), better use of network capacity (Rebain et al., 2021) and suitability to editing tasks (Liu et al., 2020). In particular, the community has recently investigated tri-planes (Chan et al., 2022), a type of hybrid representation whose discrete components are three feature planes $(xy, yz, xz)$, due to its regular grid structure and compactness. Tri-planes have been deployed for *RF* (Hu et al., 2023) and *SDF* (Wang et al., 2023).

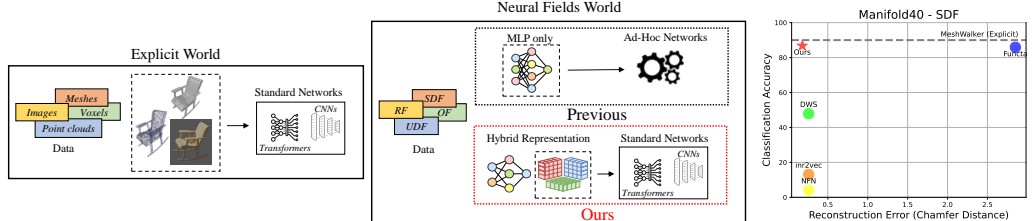

Figure 1: **Left:** Neural processing of hybrid neural fields allows us to employ well-established architectures to tackle deep learning tasks while avoiding problems related to processing MLPs, such as the high-dimensional weight space and the random initialization. **Right:** We achieve performance better than other works on this topic, close to methods that operate directly on explicit representations. without sacrificing the reconstruction quality of neural fields.

**Neural processing of neural fields.** As conjectured in De Luigi et al. (2023), due to their advantages and increasing adoption in recent years, neural fields may become one of the standard methods for storing and communicating 3D information, i.e., repositories of digital twins of real objects stored as neural networks will become available. In such a scenario, developing strategies to solve tasks such as classification or segmentation by directly processing neural fields becomes relevant to utilize these representations in practical applications. For instance, given a NeRF of a chair, classifying the weights of the MLP without rendering and processing images would be faster, less computationally demanding, and more straightforward, e.g., there is no need to understand where to sample the 3D space as *there is no sampling at all*.

Earlier methods on the topic, such as Functa (Dupont et al., 2022), approached this scenario with shared networks trained on the whole dataset conditioned on a different global embedding for each object. In this case, a neural field is realized by the shared network plus the embedding, which is then processed for downstream tasks. However, representing a whole dataset with a shared network is difficult, and the reconstruction quality of neural fields inevitably drops (see the plot in Fig. 1). For this reason, later approaches such as inr2vec (De Luigi et al., 2023), NFN (Zhou et al., 2023b), NFT (Zhou et al., 2023b), and DWSNet (Navon et al., 2023) propose to process neural fields consisting of a single large MLP, such as SIREN (Sitzmann et al., 2020), for each object. Although this strategy effectively maintains the reconstruction capabilities of neural fields, task performance suffers due to the challenges introduced by the need to handle MLPs, such as the large number of weights and the difficulty of embedding inductive biases into neural networks aimed at processing MLPs. Moreover, randomly initialized MLPs trained on the same input data can converge to drastically different regions of the weight space due to the non-convex optimization problem and the symmetries of neural weight spaces (Entezari et al., 2021; Ainsworth et al., 2023). Thus, identifying a model capable of processing MLPs and generalizing among all possible initializations is not straightforward. Previous works partially address these problems: inr2vec proposes an efficient and scalable architecture, and bypasses the initialization problem by fixing it across MLPs; NFN, NFT, and DWSNet design networks that are equivariant to weight symmetries. Nonetheless, all previous methods processing neural fields realized as single MLPs achieve unsatisfying performance, far from established architectures that operate on explicit representations, e.g., point clouds or meshes, as shown in Fig. 1 right.

**Neural processing of tri-plane neural fields.** To overcome the limitations of previous approaches and given the appealing properties of hybrid representations, in this paper, we explore the new research problem of tackling common 3D tasks by directly processing tri-plane neural fields. To this end, we analyze the information stored in the two components of this representation, which comprises a discrete feature space alongside a small MLP, and find out that the former contains rich semantic and geometric information. Based on this finding, we propose to process tri-plane neural fields by seamlessly applying, directly on the discrete feature space, standard neural architectures that have been developed and engineered over many years of research, such as CNNs (He et al., 2016) or, thanks to tri-plane compactness, even Transformers (Vaswani et al., 2017) (Fig. 1 left). Moreover, we note empirically that the same geometric structures are encoded in tri-planes fitted on the same shape from different initializations up to a permutation of the channels. Thus, we exploit this property to achieve robustness to the random initialization problem by processing tri-planes with standard architectures that are made invariant to permutation of the channels. We achieve much better

performance than all previous methods in classifying and segmenting objects represented as neural fields, almost on par with established architectures that operate on explicit representations, without sacrificing the representation quality (Fig. 1).

**Summary of our contributions.** Code available at `https://github.com/CVLAB-Unibo/triplane_processing`.
• We set forth the new research problem of solving tasks by directly processing tri-plane neural fields. We show that the discrete features encode rich semantic and geometric information, which can be elaborated by applying well-established architectures. Moreover, we note how similar information is stored in tri-planes with different initializations of the same shape. Yet, the information is organized with different channel orders.

• We show that applying well-established architectures on tri-planes achieves much better results than processing neural fields realized as a large MLP. Moreover, we reveal that employing architectures made invariant to the channel order improves performance in the challenging but more realistic scenario of randomly initialized neural fields. In this way, we almost close the gap between methods that operate on explicit representations and those working directly on neural representations.

• To validate our results, we build a comprehensive benchmark for tri-plane neural field classification. We test our method by classifying neural fields that model various fields (*UDF*, *SDF*, *OF*, *RF*). In particular, to the best of our knowledge, we are the first to classify NeRFs without explicitly reconstructing the represented signal.

• Finally, as the tri-plane structure is independent of the represented field, we train a single network to classify diverse tri-plane neural fields. Specifically, we show promising preliminary results of a unique model capable of classifying *UDF*, *SDF*, and *OF*.

## 2 RELATED WORK

**Neural fields.** Recent approaches have shown the ability of MLPs to parameterize fields representing any physical quantity of interest (Xie et al., 2021). The works focusing on representing 3D data with MLPs rely on fitting functions such as the unsigned distance (Chibane et al., 2020), the signed distance (Park et al., 2019; Gropp et al., 2020; Sitzmann et al., 2019; Jiang et al., 2020; Peng et al., 2020), the occupancy (Mescheder et al., 2019; Chen & Zhang, 2019), or the scene radiance (Mildenhall et al., 2020a). Among these approaches, SIREN (Sitzmann et al., 2020) uses periodic activation functions to capture high-frequency details. Recently, hybrid representations, in which the MLP is paired with a discrete data structure, have been introduced within the vision and graphic communities motivated by faster inference (Reiser et al., 2021), better use of network capacity (Rebain et al., 2021) and suitability to editing tasks (Liu et al., 2020). These data structures decompose the input coordinate space, either regularly, such as for voxel grids (Reiser et al., 2021; Fridovich-Keil et al., 2022; Liu et al., 2020), tri-planes (Wang et al., 2023; Chan et al., 2022; Wu & Zheng, 2022; Hu et al., 2023), and 4D tensors (Chen et al., 2022), or irregularly, such as for point clouds (Tretschk et al., 2020), and meshes (Peng et al., 2021). Unlike these works, we do not focus on designing a neural field representation, but we investigate how to directly process hybrid fields to solve tasks such as shape classification and 3D part segmentation. We focus on tri-planes due to their regular grid structure and compactness, which enable standard neural networks to process them seamlessly and effectively.

**Neural functionals.** Several recent approaches aim at processing functions parameterized as MLPs by employing other neural networks. MLPs are known to exhibit weight space symmetries (Hecht-Nielsen, 1990), i.e., hidden neurons can be permuted across layers without changing the function represented by the network. Works such as DWSNet (Navon et al., 2023), NFN (Zhou et al., 2023a), and NFT (Zhou et al., 2023b) leverage weight space symmetries as an inductive bias to develop novel architectures designed to process MLPs. Both DWSNet and NFN devise neural layers equivariant to the permutations arising in MLPs. In contrast, NFT builds upon the intuition of achieving permutation equivariance by removing positional encoding from a Transformer architecture. Among the works processing MLPs, inr2vec (De Luigi et al., 2023) is the first that focuses specifically on MLPs representing 3D neural fields. It proposes a representation learning framework that compresses neural fields of 3D shapes into embeddings, which can then be used as input for downstream tasks. In the scenario addressed by inr2vec, DWSNet, NFN, and NFT, each neural field is parameterized by its own MLP. Differently, the framework proposed in Functa (Dupont et al., 2022) relies on

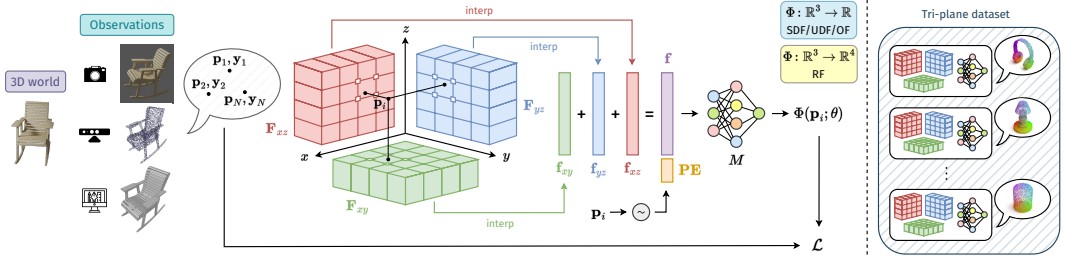

Figure 2: **Left:** Tri-plane representation and learning of each neural field. **Right:** Datasets are composed of many independent tri-plane hybrid neural fields, each representing a 3D object.

learning priors on the whole dataset with a shared network and then encoding each sample in a compact embedding. In this case, each neural field is parameterized by the shared network plus the embedding. In particular, Functa (Dupont et al., 2022) leverages meta-learning techniques to learn the shared network, which is modulated with latent vectors to represent each data point. These vectors are then used to address both discriminative and generative tasks. It is worth pointing out that, though not originally proposed as a framework to process neural fields, DeepSDF (Park et al., 2019) learns dataset priors by optimizing a reconstruction objective through a shared auto-decoder network conditioned on a shape-specific embedding. Thus, as investigated in De Luigi et al. (2023), the embeddings learnt by DeepSDF may be used for neural processing tasks similar to Functa's. However, as noted in De Luigi et al. (2023), shared network frameworks are problematic, as they cannot reconstruct the underlying signal with high fidelity and need a whole dataset to learn the neural field of an object. Thus, akin to inr2vec, DWSNet, NFN, and NFT, we adopt the setting in which an individual network represents each sample in a dataset, as it is easier to deploy in the wild and thus more likely to become the standard practice in neural field processing. Unlike all previous works, however, we process hybrid neural fields that combine an MLP with a discrete spatial data structure. By only processing the discrete component, we circumvent the issues arising from directly processing MLP weights and obtain remarkable performance.

## 3 TRI-PLANE HYBRID NEURAL FIELDS

### 3.1 PRELIMINARIES

**Neural fields.** A field is a physical quantity defined for all domain coordinates. We focus on fields describing the 3D world, and thus on $\mathbb{R}^3$ coordinates $\mathbf{p} = (x, y, z)$. We consider the 3D fields commonly used in computer vision and graphics, i.e., the *SDF* (Park et al., 2019) and *UDF* (Chibane et al., 2020), which map coordinates to the signed an unsigned distance from the closest surface, respectively, the *OF* (Mescheder et al., 2019), which computes the occupancy probability, and the *RF* (Mildenhall et al., 2020b), that outputs $(R, G, B)$ colors and density $\sigma$. A field can be modelled by a function, $\Phi$, parameterized by $\theta$. Thus, for any point $\mathbf{p}$, the field is given by $\hat{\mathbf{q}} = \Phi(\mathbf{p}; \theta)$. If parameters $\theta$ are the weights of a neural network, $\Phi$ is said to be a neural field. On the other hand, if some of the parameters are the weights of a neural network, whereas the rest encode local information within a discrete spatial structure, $\Phi$ is a hybrid neural field (Xie et al., 2021).

**Tri-plane representation.** A special case of hybrid neural fields, originally proposed in Chan et al. (2022), is parameterized by a discrete tri-plane feature map, $T$, and a small MLP network, $M$ (Fig. 2, left). $T$ consists of three orthogonal 2D feature maps, $T = (\mathbf{F}_{xy}, \mathbf{F}_{xz}, \mathbf{F}_{yz})$, with $\mathbf{F}_{xy}, \mathbf{F}_{xz}, \mathbf{F}_{yz} \in \mathbb{R}^{C \times H \times W}$, where $C$ is the number of channels and $W, H$ are the spatial dimensions of the feature maps. The feature vector associated with a 3D point, $\mathbf{p}$, is computed by projecting the point onto the three orthogonal planes so to get the 2D coordinates, $\mathbf{p}_{xy}, \mathbf{p}_{xz}$, and $\mathbf{p}_{yz}$, relative to each plane. Then, the four feature vectors corresponding to the nearest neighbours in each plane are bi-linearly interpolated to calculate three feature vectors, $\mathbf{f}_{xy}, \mathbf{f}_{xz}$, and $\mathbf{f}_{yz}$, which are summed up element-wise to obtain $\mathbf{f} = \mathbf{f}_{xy} + \mathbf{f}_{xz} + \mathbf{f}_{yz}, \mathbf{f} \in \mathbb{R}^C$. Finally, we concatenate $\mathbf{f}$ with a positional encoding (Mildenhall et al., 2020b), $\mathbf{PE}$, of the 3D point $\mathbf{p}$ and feed it to the MLP, which in turn outputs the field value at $\mathbf{p}$: $\hat{\mathbf{q}} = \Phi(\mathbf{p}; \theta) = M([\mathbf{f}, \mathbf{PE}])$. We implement $M$ with *sin* activation functions (Sitzmann et al., 2020) to better capture high-frequency details.

| Method | Type | # Params (K) | Mesh from *SDF* | | Point Cloud from *UDF* | |
|---|---|---|---|---|---|---|
| | | | CD (mm) | F-score (%) | CD (mm) | F-score (%) |
| inr2vec (De Luigi et al., 2023) | Single | 800 | 0.26 | 69.7 | 0.21 | 65.5 |
| Tri-plane | Single | 64 | 0.18 | 68.6 | 0.24 | 60.7 |
| Tri-plane | Shared | 64 | 1.57 | 42.9 | 3.45 | 33.3 |
| DeepSDF (Park et al., 2019) | Shared | 2400 | 6.6 | 25.1 | 5.6 | 5.7 |
| Functa (Dupont et al., 2022) | Shared | 7091 | 2.85 | 21.3 | 12.8 | 5.8 |

Table 1: **Results of mesh and point cloud reconstruction on the Manifold40 test set.** "Single" and "Shared" indicate neural fields trained on each shape independently or on the whole dataset.

**Learning tri-planes.** To learn a field, we optimize a $(T, M)$ pair *for each 3D object*, starting from randomly initialized parameters, $\theta$, for both $M$ and $T$. We sample $N$ points $\mathbf{p}_i$ and feed them to $T$ and $M$ to compute the corresponding field quantities $\hat{\mathbf{q}}_i = \Phi(\mathbf{p}_i; \theta)$. Then, we optimize $\theta$ with a loss, $\mathcal{L}$, capturing the discrepancy between the predicted fields $\hat{\mathbf{q}}_i$ and the ground truth $\mathbf{y}_i$, applying an optional mapping between the output and the available supervision if needed (e.g., volumetric rendering in case of *RF*). An overview of this procedure is shown on the left of Fig. 2 and described in detail in Appendix A. We repeat this process for each 3D shape of a dataset, thereby creating a dataset of tri-plane hybrid neural fields (Fig. 2, right). We set $C$ to 16 and both $H$ and $W$ to 32. We use MLPs with three hidden layers, each having 64 neurons. We note that our proposal is independent of the learning procedure, and, in a scenario in which neural fields are a standard 3D data representation, we would already have datasets available.

## 3.2 TRI-PLANE ANALYSIS

We investigate here the benefits of tri-planes for 3D data representation and neural processing. Firstly, we assess their reconstruction capability, which is crucial in a world where neural fields may be used as a standard way to represent 3D assets. Secondly, we analyze the information learned in the 2D planes and how to be robust to the random initialization problem when handling tri-planes.

**Reconstruction quality.** We assess the tri-plane reconstruction performance by following the benchmark introduced in De Luigi et al. (2023). In Table 1, we present the quantitative outcomes obtained by fitting *SDF*s and *UDF*s from meshes and point clouds of the Manifold40 dataset (Hu et al., 2022). We compare with neural fields employed in inr2vec (De Luigi et al., 2023) and alternatives based on a shared architecture, such as DeepSDF (Park et al., 2019) and Functa (Dupont et al., 2022). Given the *SDF* and *UDF* fields learned by each framework, we reconstruct the explicit meshes and point clouds as described in Appendix B.1 and evaluate them against the ground-truths. To conduct this evaluation, we sample dense point clouds of 16,384 points from both the reconstructed and ground-truth shapes. We employ the Chamfer Distance (Fan et al., 2017) and the F-Score (Tatarchenko et al., 2019) to evaluate fidelity to ground-truths. As for meshes, the tri-planes representation stands out with the lowest Chamfer Distance (CD) (0.18 mm), indicating its excellent reconstruction quality despite the relatively small number of parameters (only 64K). For point clouds, tri-planes produce reconstructions slightly worse than inr2vec but still comparable, i.e., 0.21m vs 0.24mm CD. In Appendix B.2 (Fig. 10), we show reconstructions attained from tri-plane representations for various types of fields. Moreover, in agreement with the findings of De Luigi et al. (2023), Table 1 shows that shared network frameworks such as DeepSDF and Functa yield significantly worse performance in terms of reconstruction quality. We finally point out how sharing the MLP for all tri-planes is not as effective as learning individual neural fields (third vs second row). These results support our intuition that reconstruction quality mandates hybrid neural fields optimized individually on each data sample and highlight the importance of investigating the direct neural processing of these representations. In Appendix B.3 (Fig. 5, Fig. 6), we show the reconstructions obtained by tri-planes and the other approaches considered in our evaluation.

**Tri-plane content.** To investigate how to directly process tri-plane neural fields, we inspected the content of their discrete spatial structure by visualizing the features stored in a plane alongside the view of the object rendered from the vantage point corresponding to the plane. Examples of these visualizations are depicted in Fig. 3 (left) for various objects such as a car, an airplane, and a bottle. To visualize features as a single image, displayed by a *viridis* colormap, we take a sum across the feature channels at each spatial location. These visualizations show clearly that the tri-plane spatial structure learns the object shape, i.e., it contains information about its geometry. For this reason

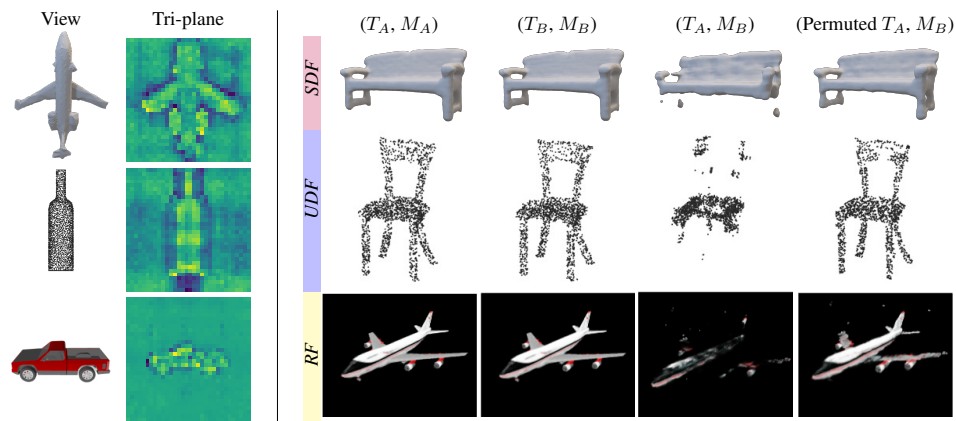

Figure 3: **Left:** For three different hybrid neural fields (from top to bottom: *SDF*, *UDF*, *RF*) we render a view of the reconstructed 3D object alongside the corresponding tri-plane feature map. **Right:** From left to right, reconstructions of two (tri-plane, MLP) pairs with different initializations, namely $(T_A, M_A)$ and $(T_B, M_B)$; the mixed pair $(T_A, M_B)$; a channel permutation of $T_A$ and $M_B$.

and further investigations reported in Appendix D, we conjecture, and demonstrate empirically in subsequent sections, that to tackle tasks such as classification and segmentation we can discard the MLPs and process only the tri-plane structure of the neural fields. Remarkably, the regular grid structure of tri-planes allows us to deploy popular and effective neural architectures, such as CNNs and Transformers. On the contrary, direct ingestion of MLPs for neural processing is problematic and leads to sub-optimal task performance.

**Random initialization.** Furthermore, we investigate the effect of random initializations on tri-plane neural fields. We note here that by "random initialization" we mean that each (tri-plane, MLP) pair adheres to the same initialization scheme but has a different random seed (see Appendix E.5). We find out empirically that the main difference between tri-plane structures learnt from different optimizations of the same shape lies in the channel order within a feature plane. Indeed, we conducted experiments where we fit the same 3D shape twice (see Fig. 3 (right)), starting from two random initializations of both the tri-plane structure and the MLP weights. Although the geometric content of the two tri-planes is similar, due to the different initialization, the tri-plane learnt in the first run cannot be used with the MLP obtained in the second (third column of Fig. 3, right side), and vice-versa. However, it is always possible to find a suitable permutation of the channels of the first tri-plane such that the second MLP can correctly decode its features (fourth column of Fig. 3, right side), and vice-versa. We found the right permutation by a brute-force search based on maximizing reconstruction quality. To make the search feasible, we used a smaller number of channels, i.e., $C = 8$ rather than $C = 16$. Still, the experimental results in Section 4.1 support our belief that the main source of variance across randomly initialized tri-plane optimizations of the same shape consists of a permutation of the channel order. Thus, unlike neural fields realized as MLPs, with tri-planes, it is straightforward to counteract the nuisances due to random initialization by adopting standard architectures made invariant to the channel order.

### 3.3 ARCHITECTURES FOR NEURAL PROCESSING OF TRI-PLANE NEURAL FIELDS

Based on the above analysis, we propose to process tri-planes with Transformers (Vaswani et al., 2017). In particular, we propose to rely on a Transformer encoder without positional encoding, which is equivariant to token positions. By tokenizing tri-planes so that each token represents a channel of a plane, such architecture seamlessly computes representations equivariant to the order of the channels. Specifically, we unroll each channel of size $H \times W$, to obtain a token of dimension $HW$ within a sequence of length $3C$ tokens. These tokens are then linearly projected and fed into the Transformer. The output of the encoder is once again a sequence of $3C$ tokens.

For global tasks like classification, the output sequence is subsequently subjected to a max pool operator to obtain a global embedding that characterizes the input shape. In our experiments, this embedding is then processed through a stack of fully connected layers to compute the logits. The

| Method | Type | Input | UDF | | | SDF | OF | RF |
|---|---|---|---|---|---|---|---|---|
| | | | ModelNet40 | ShapeNet10 | ScanNet10 | Manifold40 | ShapeNet10 | ShapeNetRender |
| DeepSDF (Park et al., 2019) | Shared | Latent vector | 41.2 | 76.9 | 51.2 | 64.9 | – | – |
| Functa (Dupont et al., 2022) | Shared | Modulation | **87.3** | 83.4 | 56.4 | 85.9 | 36.3 | – |
| inr2vec (De Luigi et al., 2023) | Single | MLP | 10.6 | 42.0 | 40.9 | 13.1 | 38.6 | – |
| MLP | Single | MLP | 3.7 | 28.8 | 36.7 | 4.2 | 29.6 | 22.0 |
| NFN (Zhou et al., 2023a) | Single | MLP | 9.0 | 9.0 | 45.3 | 4.1 | 33.8 | 87.0 |
| NFT (Zhou et al., 2023b) | Single | MLP | 6.9 | 6.9 | 45.3 | 4.1 | 33.8 | 85.3 |
| DWSNet (Navon et al., 2023) | Single | MLP | 56.3 | 78.4 | 62.2 | 47.9 | 79.1 | 83.1 |
| Ours | Single | Tri-plane | 87.0 | **94.1** | **69.1** | **86.8** | **91.8** | **92.6** |

Table 2: **Test set accuracy for shape classification across neural fields.** We compare several frameworks capable of processing neural fields.

way the tokens are defined, the absence of positional encoding, and the final max pool operator allow for achieving invariance to the channel order. For dense tasks like part segmentation, we also utilize the decoder part of Transformers. More specifically, we treat the coordinates queries to segment as a sequence of input tokens to the decoder. Each point $\mathbf{p}$ with coordinates $(x, y, z)$ undergoes positional encoding (Mildenhall et al., 2020b) and is then projected to a higher-dimensional space using a linear layer. By leveraging the cross-attention mechanisms within the decoder, each input token representing a query point can globally attend to the most relevant parts of the tri-planes processed by the encoder to produce its logits. Additional details on the architectures are reported in Appendix E.3.

# 4    TASKS ON NEURAL FIELDS

## 4.1    NEURAL FIELD CLASSIFICATION

**Benchmark.** We perform extensive tests to validate our approach. In so doing, we build the first neural field classification benchmark, where we compare all the existing proposals for neural field processing on the task of predicting the category of the objects represented within the field without recreating the explicit signal. Specifically, we test all methods on *UDF* fields obtained from point clouds of ModelNet40 (Wu et al., 2015), ShapeNet10 (Qin et al., 2019), and ScanNet10 (Qin et al., 2019); *SDF* fields learned from meshes of Manifold40 (Hu et al., 2022); *OF* fields obtained from voxels grids of ShapeNet10. In addition, we provide for the first time classification results on neural radiance fields (*RF*), learned from ShapenetRender (Xu et al., 2019). See Appendix E.1 for more details on the benchmark. Besides a simple MLP baseline, we compare with frameworks designed to process neural fields realized as MLPs, i.e., inr2vec (De Luigi et al., 2023), NFN (Zhou et al., 2023a), NFT (Zhou et al., 2023b), and DWSNet (Navon et al., 2023). These methods process single MLP neural fields, which we implement as SIREN networks (Sitzmann et al., 2020). Differently from De Luigi et al. (2023), the MLPs in our benchmark are *randomly initialized* to simulate real-world scenarios. Unlike all previous methods, ours processes individual tri-plane neural fields, which are also randomly initialized. Moreover, we compare with frameworks where neural fields are realized by a shared network and a small latent vector or modulation, i.e., DeepSDF (Park et al., 2019) and Functa (Dupont et al., 2022). Whenever possible, we use the official code released by the authors to run the experiments. Note that not all frameworks can be easily extended to all fields. Therefore, we only test each framework in the settings that are compatible with our resources and that do not require fundamental changes to the original implementations (see Appendix E.2 for more details).

**Results.** As we can observe in Table 2, overall, shared architecture frameworks (DeepSDF and Functa) outperform previous methods that directly operate on neural fields represented as a single neural network. However, we point out again that the reconstruction capability of such frameworks is poor, as shown in Section 3.2. Conversely, previous methods that utilize individual neural fields demonstrate superior reconstruction quality but struggle to perform effectively in real-world scenarios where shapes need to be fitted starting from arbitrary initialization points. inr2vec makes the assumption of learning all MLPs starting from the same initialization, and it does not work when this initialization schema is not applied. Among the family of methods that adopt layers equivariant and invariant to permutations of the neurons, only DWSNet works on the large MLPs constituting our benchmark, though performance tends to be worse than shared network approaches. Our method delivers the best of both worlds: it ingests tri-planes neural fields, which exhibit excellent reconstruction quality while achieving the best performance overall, often surpassing by a large margin all other methods, including those relying on a shared neural field, e.g., the accuracy on ScanNet10 is 56.4 for Functa vs 69.1 for our method. Hence, we can state confidently that our approach achieves the best trade-off

| Method | Input | ModelNet40 | ShapeNet10 | ScanNet10 | Manifold40 | ShapeNet10 | ShapeNetRender |
|---|---|---|---|---|---|---|---|
| Ours | Tri-plane | 87.0 | 94.1 | 69.1 | 86.8 | 91.8 | 92.6 |
| PointNet (Qi et al., 2017a) | Point Cloud | 88.8 | 94.3 | 72.7 | – | – | – |
| MeshWalker (Lahav & Tal, 2020) | Mesh | – | – | – | 90.0 | – | – |
| Conv3DNet (Maturana & Scherer, 2015) | Voxel | – | – | – | – | 92.1 | – |
| ResNet50 (He et al., 2016) | Images | – | – | – | – | – | 94.0 |

Table 3: **Comparison with explicit representations. Top:** Test set accuracy of our neural field processing method. **Bottom:** Standard networks trained and tested on explicit representations.

between classification accuracy and reconstruction quality. Finally, we highlight that our proposal is effective with all the datasets and kinds of fields addressed in the experiments.

**Comparison with explicit representations.** In Table 3, we compare our method against established architectures specifically designed to process explicit representations. For a fair comparison, we reconstruct the explicit data from each field so that exactly the same shapes are used in each experiment. Practically, we reconstruct point clouds, mesh, and voxel grids from *UDF*, *SDF*, and *OF*, respectively. Then, we process them with specialized architectures, i.e., PointNet (Qi et al., 2017a) for point clouds, MeshWalker (Lahav & Tal, 2020) for meshes, and Conv3DNet (Maturana & Scherer, 2015) for voxel grids. As for *RF*, we render a multi-view dataset with 36 views for each object. Then, we train 36 per-view ResNet50 (He et al., 2016) so as to ensemble the predictions at test time. We highlight how our proposal, which can classify every neural field with the same standard architecture, almost closes the performance gap with respect to *specialized* architectures designed to process explicit representations. Noticeably, we show that NeRFs can be classified accurately from the features stored in a tri-plane structure without rendering any images.

**Towards universal tri-plane classification.** Finally, to the best of our knowledge, we implement for the first time a *universal tri-plane classifier*, i.e., a model which can be trained and tested with any kind of tri-plane hybrid neural field. Indeed, since the tri-plane structure, as well as the neural processing architecture, are just the same, regardless of the kind of field, we can seamlessly learn a unified model able to classify a variety of fields. For example, we start from the meshes of the Manifold40 dataset and obtain the corresponding point clouds and voxel grids so as to fit three different fields (*SDF*, *UDF*, and *OF*). Accordingly,

| Train | | | Test | | |
|---|---|---|---|---|---|
| *UDF* | *SDF* | *OF* | *UDF* | *SDF* | *OF* |
| ✓ | | | 84.7 | 78.4 | 15.6 |
| | ✓ | | 67.3 | 86.8 | 11.9 |
| | | ✓ | 49.3 | 46.9 | 77.7 |
| ✓ | ✓ | ✓ | **87.4** | **87.8** | **80.3** |

Table 4: **Universal tri-plane classifier.** Test set accuracy on Manifold40.

we build training, validation, and test sets with samples drawn from all three fields. More precisely, if a shape appears in a set represented as an *SDF*, it also appears in that set as a *UDF* and *OF*. Then, as reported in Table 4, we run classification experiments by training models on each of the individual fields as well as on all three of them jointly. The results show that when a classifier is trained on only one field, it may not generalize well to others. On the other hand, a single model trained jointly on all fields not only works well with test samples coming from each one, but it also outperforms the models trained individually on a single kind of field.

## 4.2 Neural field 3D part segmentation

We explore here the potential of our method in tackling dense prediction tasks like part segmentation, where the goal is to predict the correct part label for any given 3D point. In Table 5, we compare our method to inr2vec (De Luigi et al., 2023), which was trained on fields generated from random initialization and is the only competitor capable of addressing the part segmentation task. Our experiments were conducted by fitting *UDF* fields from point clouds of 2048 points from the ShapeNetPart dataset (Yi et al., 2016). As a reference, we present the results obtained using specialized architectures commonly used for point cloud segmentation, like PointNet, PointNet++, and DGCNN. Akin to De Luigi et al. (2023), all models are trained on the point clouds reconstructed from the fitted fields. We observe that our proposal outperforms inr2vec by a large margin, with improvements of 20% and 16.7% for instance and class mIoU, respectively. Moreover, Table 5 demonstrates once again that tri-planes are effective in substantially reducing the performance gap between processing neural fields and explicit representations.

| Method | Input | instance mIoU | class mIoU | airplane | bag | cap | car | chair | earphone | guitar | knife | lamp | laptop | motor | mug | pistol | rocket | skateboard | table |
|---|---|---|---|---|---|---|---|---|---|---|---|---|---|---|---|---|---|---|---|
| inr2vec (De Luigi et al., 2023) | MLP | 64.2 | 64.5 | 57.9 | 72.9 | 67.8 | 56.4 | 67.6 | 48.4 | 81.6 | 70.6 | 55.5 | 88.8 | 51.5 | 87.2 | 64.7 | 40.1 | 58.4 | 62.5 |
| Ours | Tri-plane | **84.2** | **81.3** | **83.0** | **80.2** | **87.4** | **76.6** | **90.2** | **68.2** | **91.6** | **85.9** | **82.1** | **95.0** | **70.7** | **94.4** | **81.9** | **59.0** | **73.4** | **80.9** |
| PointNet (Qi et al., 2017a) | Point Cloud | 83.1 | 78.96 | 81.3 | 76.9 | 79.6 | 71.4 | 89.4 | 67.0 | 91.2 | 80.5 | 80.0 | 95.1 | 66.3 | 91.3 | 80.6 | 57.8 | 73.6 | 81.5 |
| PointNet++ (Qi et al., 2017b) | Point Cloud | 84.9 | 82.73 | 82.2 | 88.8 | 84.0 | 76.0 | 90.4 | 80.6 | 91.8 | 84.9 | 84.4 | 94.9 | 72.2 | 94.7 | 81.3 | 61.1 | 74.1 | 82.3 |
| DGCNN (Wang et al., 2019) | Point Cloud | 83.6 | 80.86 | 80.7 | 84.3 | 82.8 | 74.8 | 89.0 | 81.2 | 90.1 | 86.4 | 84.0 | 95.4 | 59.3 | 92.8 | 77.8 | 62.5 | 71.6 | 81.1 |

Table 5: **Part segmentation results. Top:** Implicit frameworks. **Bottom:** Methods on explicit representation. In **bold**, best results among frameworks processing neural fields.

| Method | Input | UDF | | | SDF | OF |
|---|---|---|---|---|---|---|
| | | ModelNet40 | ShapeNet10 | ScanNet10 | Manifold40 | ShapeNet10 |
| MLP | Tri-plane | 41.6 | 84.2 | 55.8 | 40.2 | 79.1 |
| CNN | Tri-plane | 82.2 | 92.1 | 63.4 | 82.5 | 88.4 |
| PointNet | Tri-plane | 85.8 | 93.4 | **69.3** | 85.6 | 91.5 |
| Spatial PointNet | Tri-plane | 32.3 | 65.4 | 51.3 | 37.0 | 54.7 |
| Transformer | Tri-plane | **87.0** | **94.1** | 69.1 | **86.8** | **91.8** |

Table 6: **Ablation study of architectures for tri-plane neural field classification**

## 4.3 DIFFERENT ARCHITECTURES FOR TRI-PLANE PROCESSING

In Table 6, we compare several plausible alternatives to Transformers for processing tri-planes, which have roughly the same number of parameters and have been trained with the same hyperparameters. As discussed previously, since tri-planes contain an informative and regular discrete data structure and are compact, they can be processed with standard architectures. Hence, we test an MLP, a ResNet50 (He et al., 2016), and two variants of PointNet all with roughly the same parameters. A simple MLP that processes the flattened tri-planes (row 1) severely under-performs with respect to the alternatives, likely due to its inability to capture the spatial structures present in the input as well as its sensitivity to the channel permutation caused by random initializations. A standard CNN like ResNet50, processing tri-planes stacked together and treated as a multi-channel image of resolution $W \times H$, is instead equipped with the inductive biases needed to effectively process the spatial information contained in the tri-planes (Fig. 3) and already delivers promising performance, although it cannot cope with channel permutations. The two variants of PointNet show the importance of invariance to channel order. In the first variant (row 3), each channel is flattened to create a set of vectors in $\mathbb{R}^{W \times H}$ with $3C$ elements, and then the max pool operator is applied to extract a global embedding invariant to the channel order that is fed to a classifier. We observe here better performance across all fields than those attained by CNN. If we instead unroll tri-planes along the channel dimension to create a set of vectors in $\mathbb{R}^{3C}$ with $W \times H$ elements (row 4), the results are poor, as this arrangement does not make the network invariant to the channel order but to the spatial position of the features. Finally, we report (row 5) results for the Transformer architecture adopted in this paper, which, similarly to the previous PointNet, is invariant to the channel order thanks to the max-pool operator and yields slightly better performance, probably due to the attention mechanism that better captures inter-channel correlations.

## 5 CONCLUDING REMARKS AND LIMITATIONS

We have shown that tri-plane hybrid neural fields are particularly amenable to direct neural processing without sacrificing representation quality. Indeed, by feeding only the tri-plane structure into standard architectures, such as Transformers, we achieve better classification and segmentation performance compared to previous frameworks aimed at processing neural fields and dramatically shrink the gap with respect to specialized architectures designed to process 3D data represented explicitly. To validate our intuitions, we propose the first benchmark for neural processing of neural fields, which includes the main kinds of fields used to model the 3D world as well as all the published methods that tackle this very novel research problem. Within our experimental evaluation, we show for the first time that NeRFs can be effectively classified without rendering any images. A major limitation of our work is that tri-plane neural fields are specific to 3D world modeling. Thus, we plan to address other kinds of hybrid neural fields, like those relying on sparse feature sets (Li et al., 2022) as well as other kinds of signals, such as time-varying radiance fields (Sara Fridovich-Keil and Giacomo Meanti et al., 2023). Other research directions deal with processing hybrid neural fields capturing large scenes featuring multiple objects to address tasks like 3D object detection or semantic segmentation.

## ACKNOWLEDGEMENT

We wish to thank Iacopo Curti for the results produced during his master's thesis.

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

# A    LEARNING TRI-PLANE NEURAL FIELDS

In this section, we outline the procedure used to learn a single (tri-plane, MLP) pair, denoted as $(T, M)$, to create the datasets of hybrid neural fields. We note that, nonetheless, our proposal is agnostic to how neural fields were trained, as, in the scenario we consider, neural fields would be available as a standard data representation and ready to be processed.

To learn a field, we optimize the parameters $\theta$ of both $T$ and $M$ with a loss between the reconstructed field $\hat{\mathbf{q}}_i$ and the sensor measurement $\mathbf{y}_i$. The optimized weights $\theta^*$ are obtained as follows:

$$\theta^* = \arg\min_\theta \frac{1}{N} \sum_{i=1}^{N} \mathcal{L}(\alpha_1(\mathbf{y}_i), \alpha_2(\Phi_\theta(\mathbf{p}_i)))$$

where $\Phi : \mathbb{R}^3 \rightarrow \mathbb{R}^d$ represents the field function defined by both the tri-plane and the MLP, and $\mathcal{L}$ is a function that computes the error between predicted and ground-truth values, with $d = 1$ when supervising the fitting process of *UDF*, *SDF*, or *OF*, while $d = 4$ for *RF*. $\alpha_1$ represents the mapping from the sensor domain to the field, e.g., from a mesh to its *SDF*, which does not need to be differentiable. On the other hand, $\alpha_2$ represents a *forward map* between the output of the field and the domain of the available supervisory signal and must be a differentiable function (e.g., $\alpha_2$ models volumetric rendering for *RF*). $\alpha_1$ and $\alpha_2$ can also be identity functions, e.g., this is the case for $\alpha_2$ when learning an *SDF* from a mesh, as supervise directly with the field values. In the remainder of this section, we describe the steps required to create our neural field datasets.

***UDF* from a point cloud.** Given a point $\mathbf{p} \in \mathbb{R}^3$, *UDF*$(\mathbf{p})$ is defined as $\min_{\mathbf{r} \in \mathcal{P}} \|\mathbf{p} - \mathbf{r}\|_2$, namely the Euclidean distance from $\mathbf{p}$ to the closest point $\mathbf{r}$ of the point cloud. For each shape, we first sample 600K points and compute the corresponding ground truth *UDF* values $q_i$. Between these points, 100K are sampled on the surface, 250K close to the surface, 200K points at a medium distance from the surface, 25K far from the surface, and an additional 25K scattered uniformly in the volume. More precisely, close points are computed by corrupting the point on the surface with noise sampled from the normal distribution $\mathcal{N}(0, 0.001)$, medium-distance ones with noise from $\mathcal{N}(0, 0.01)$, and far-away ones with noise from $\mathcal{N}(0, 0.1)$. Then, during the optimization process, we randomly select $N = 50K$ points for each batch, apply the interpolation scheme explained in Section 3.1, retrieve the correct feature vector from the tri-plane, concatenate it with the positional encoding of $\mathbf{p}$, and feed it to the MLP to compute the loss. This procedure is repeated 1000 times. As for the training objective, we follow De Luigi et al. (2023), i.e., we scale the *UDF* ground truth labels into the $[0, 1]$ range, with 0 and 1 representing the maximum and minimum distance from the surface, respectively. We then constrain predictions to be in that range through a final sigmoid activation function. Finally, we optimize the weights of $\Phi$ using the binary cross entropy between the scaled ground truth labels $q_i$ and the predicted field values $\hat{q}_i$:

$$\mathcal{L}_{\text{bce}} = -\frac{1}{N} \sum_{i=1}^{N} q_i \log(\hat{q}_i) + (1 - q_i) \log(1 - \hat{q}_i) \tag{1}$$

***SDF* from a mesh.** Given a point $\mathbf{p} \in \mathbb{R}^3$, *SDF*$(\mathbf{p})$ is defined as the Euclidean distance from $\mathbf{p}$ to the closest point of the surface, with positive sign if $\mathbf{p}$ is outside the shape and negative sign otherwise. For watertight meshes, we can easily understand whether a point lies inside or outside the surface by analyzing normals. We compute *SDF* ground truth values $s_i$ for 600K sampled points. Then, we compute the binary cross entropy loss of Eq. (1) on $N = 50K$ points. We found 600 optimization steps to be enough to achieve satisfying reconstruction quality. Similarly to the *UDF*, the *SDF* values $s_i$ are scaled into the $[0, 1]$ range, with 0 and 1 representing the maximum and minimum distance from the surface, respectively and 0.5 representing the surface level set.

***OF* from a voxel grid.** Given a point $\mathbf{p} \in \mathbb{R}^3$, *OF*$(\mathbf{p})$ is defined as the probability $o_i$ of $\mathbf{p}$ being occupied. For voxel grids, the fitting process is straightforward, as each cube $c_i$ can either be occupied (value 1) or empty (value 0). Thus, we can directly apply the binary cross-entropy loss. We use the same protocol applied for point clouds and optimize for 1000 steps while sampling 50K points at each step. However, due to the high imbalance between empty and full cells, we follow De Luigi et al. (2023) and employ a focal loss (Lin et al., 2017):

$$\mathcal{L}_{\text{focal}} = -\frac{1}{N} \sum_{i=1}^{N} \beta(1 - o_i)^\gamma c_i \log(o_i) + (1 - \beta)o_i^\gamma (1 - c_i) \log(1 - o_i)$$

with $\beta$ and $\gamma$ representing the balancing and focusing parameters, respectively.

**RF from images.** Given a point $\mathbf{p} \in \mathbb{R}^3$, *RF*$(\mathbf{p})$ (Mildenhall et al., 2020b) is a 4-dimensional vector containing the $(R, G, B)$ color channels and density $\sigma$ of the point. To fit a *RF*, we rely on the NerfAcc library (Li et al., 2023). Specifically, we implement a simplified version of NeRF that does not take into account the viewing direction and consists of a (tri-plane, MLP) pair that predicts the four field values. To render RGB images, we leverage the volumetric rendering implementation of Li et al. (2023). In particular, at each iteration, for a given camera pose, a batch of rays (128 in our case) is selected with the corresponding RGB ground truth values and 3D points are sampled along these rays. The coordinates of these points are then interpolated, as described in Section 3.1, to extract features from the tri-plane and fed to the MLP, which in turn predicts the $(R, G, B, \sigma)$ values for each of them. Finally, by means of volumetric rendering, a final RGB vector color is obtained and compared to the ground truth via a smooth L1 loss. Each *RF* is trained for 1500 steps.

# B    EXPLICIT RECONSTRUCTION FROM NEURAL FIELDS

In this section, we discuss how to sample 3D explicit representations from neural fields and show some examples of such reconstructions.

## B.1    SAMPLING EXPLICIT REPRESENTATIONS

**Point cloud from *UDF*.** We generate a point cloud based on the corresponding *UDF* using a slightly adapted version of the algorithm introduced by Chibane et al. (2020). The fundamental concept involves querying the *UDF* with points distributed throughout the specific region of the 3D space under consideration. These points are then projected onto the isosurface based on their predicted *UDF* values. For a given point $\mathbf{p} \in \mathbb{R}^3$, its updated position $\mathbf{p}_{\text{new}}$ is determined as follows:

$$\mathbf{p}_{\text{new}} = \mathbf{p} - \Phi(\mathbf{p}; \theta) \frac{\nabla_{\mathbf{p}} \Phi(\mathbf{p}; \theta)}{\|\nabla_{\mathbf{p}} \Phi(\mathbf{p}; \theta)\|} \tag{2}$$

where $\Phi(\mathbf{p}; \theta)$ represents the field value at point $\mathbf{p}$ which is approximated using the $(T, M)$ pair with parameters $\theta$. It is important to note that the negative gradient of the *UDF* indicates the direction of the steepest decrease in distance from the surface, effectively pointing towards the nearest point on the isosurface. Eq. (2) can thus be interpreted as shifting point $\mathbf{p}$ along the direction of maximum *UDF* decrease, ultimately arriving at point $\mathbf{p}_{\text{new}}$ on the surface. However, it is crucial to observe that $\Phi$ serves as an approximation of the true *UDF*. This raises two key considerations: (i) the gradient of $\Phi$ must be normalized, as illustrated in Eq. (2), whereas the gradient of the actual *UDF* maintains norm 1 everywhere except on the surface; (ii) the predicted *UDF* value can be imprecise, potentially resulting in point $\mathbf{p}$ remaining distant from the surface even after applying Eq. (2). To address the second point, we refine $\mathbf{p}_{\text{new}}$ by iteratively repeating the update described in Eq. (2). With each iteration, the point gradually approaches the surface, where the values approximated by $\Phi$ become more accurate, eventually placing the point precisely on the isosurface. The whole algorithm for sampling a dense point cloud from a given *UDF* entails the following steps: (i) generate a set of points uniformly scattered within the specified region of 3D space and predict their *UDF* values using the provided $(T, M)$ pair; (ii) discard points with predicted *UDF* values exceeding a fixed threshold (0.05 in our experiments). For the remaining points, update their coordinates iteratively using Eq. (2), typically requiring 5 updates for satisfactory results; (iii) repeat the entire procedure until the reconstructed point cloud reaches the desired number of points.

**Triangle mesh from *SDF*.** We employ the Marching Cubes algorithm (Lorensen & Cline, 1987) to construct a mesh based on the corresponding *SDF*. The Marching Cubes process involves systematically traversing the 3D space by evaluating the *SDF* at 8 locations simultaneously, forming the vertices of a tiny virtual cube. This traversal continues until the entirety of the desired 3D region has been covered. For each cube, the algorithm identifies the triangles necessary to represent the portion of the isosurface passing through it. These triangles from all cubes are then integrated to create the final reconstructed surface. To determine the number and placement of triangles for an individual cube, the algorithm examines the *SDF* values at pairs of neighbouring vertices within the cube. A triangle vertex is inserted between two vertices with opposing *SDF* signs. Because the possible combinations of *SDF* signs at cube vertices are limited, a lookup table is generated to retrieve the triangle configuration for a given cube. This configuration is derived from the *SDF* signs at the eight

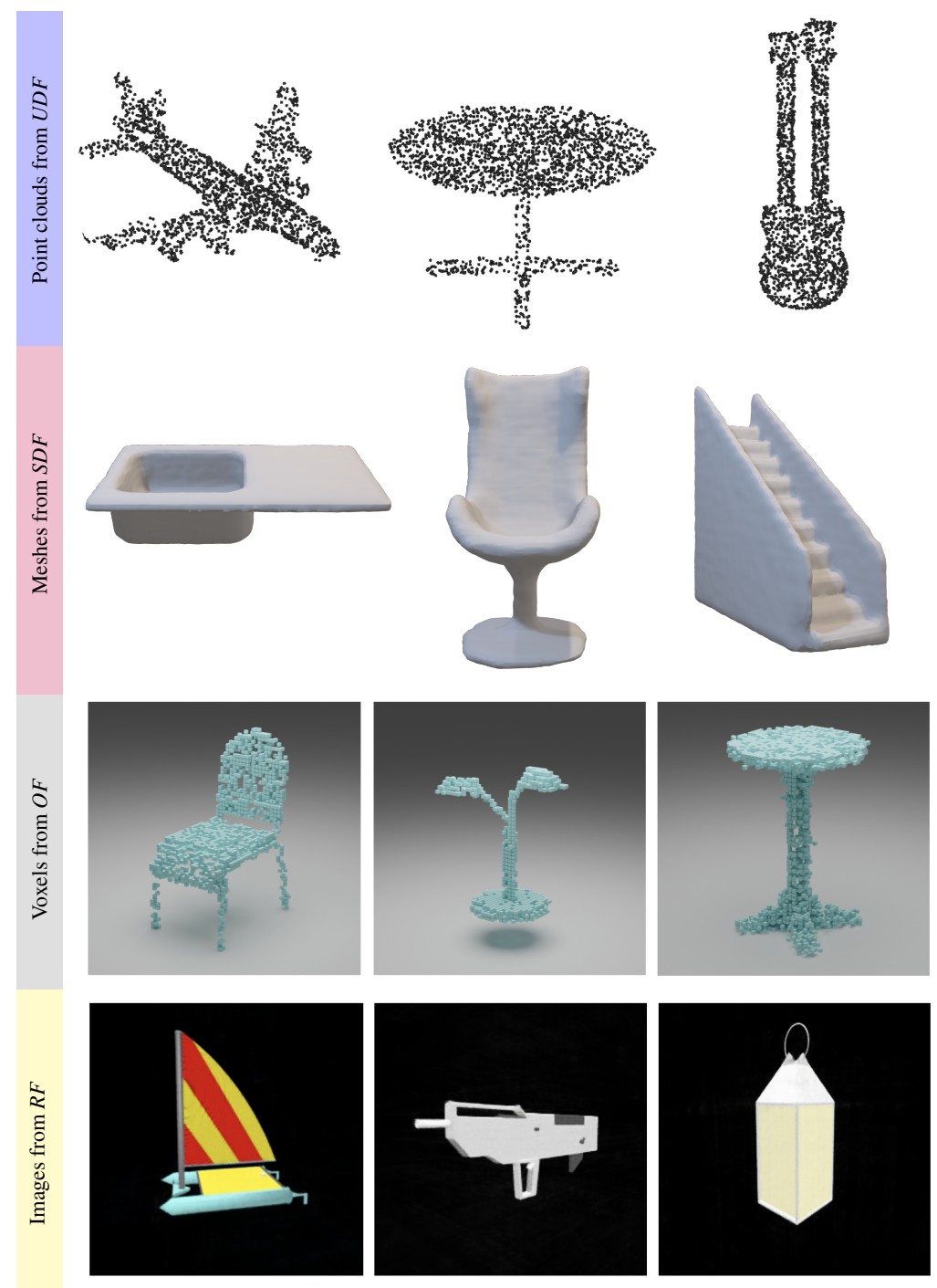

Figure 4: **Tri-plane reconstruction examples of point clouds from *UDF*, meshes from *SDF*, voxels from *OF*, and images from *RF*** (from top to bottom)

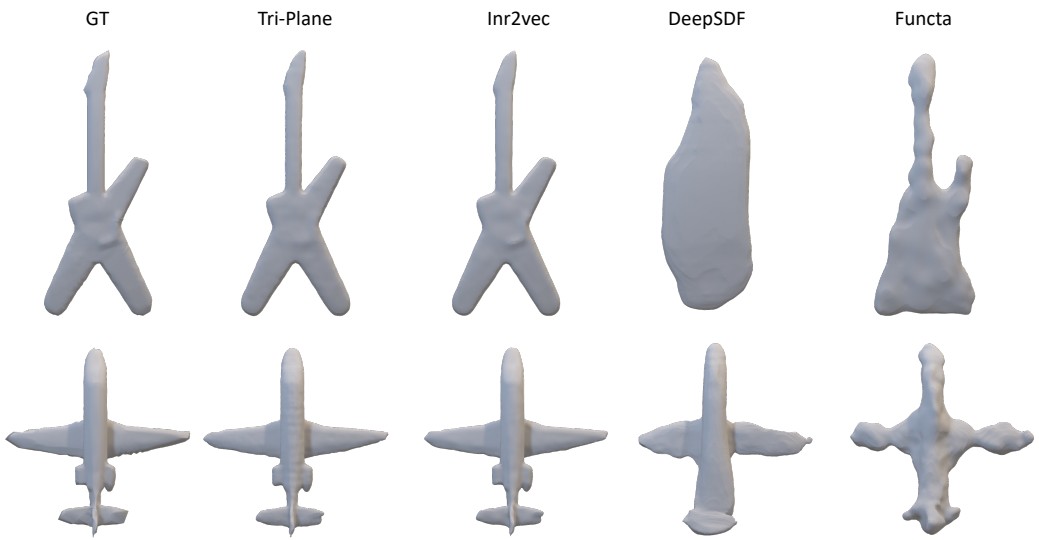

Figure 5: **Reconstruction comparison for Manifold40 meshes obtained from *SDF***

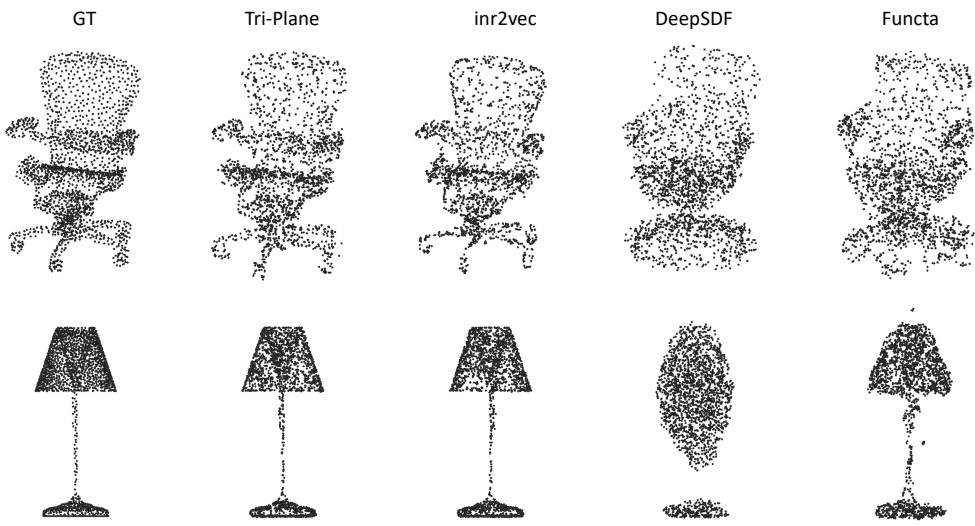

Figure 6: **Reconstruction comparison for ModelNet40 point clouds obtained from *UDF***

vertices of the cube, which are combined into an 8-bit integer and used as a lookup table key. Once the triangle configuration for a cube is retrieved, the vertices of the triangles are positioned along the edges connecting the cube vertices. This positioning is accomplished through linear interpolation of the two *SDF* values associated with each edge.

**Voxel grid from *OF*.** To generate a voxel grid from its *OF*, we employ a straightforward procedure. Each neural field is trained to estimate the likelihood of a specific voxel being full when provided with the 3D coordinates of the voxel center. Consequently, the initial step in reconstructing the fitted voxels involves constructing a grid with the desired resolution, denoted as $V$. Then, the field is queried using the $V^3$ centroids of this grid, and it produces an estimated probability of occupancy for each centroid. Eventually, we designate voxels as occupied only if their predicted probability surpasses a predefined threshold, which we have empirically set to 0.4. This threshold selection has been determined through

| Method | Type | # Params (K) | Mesh from *SDF* | |
| | | | CD (mm) | F-score (%) |
| --- | --- | --- | --- | --- |
| Voxel | Single | 554 | 0.19 | 69.1 |
| Tri-plane | Single | 64 | 0.18 | 68.6 |

Table 7: **Mesh reconstruction results on the Manifold40 test set (voxel grid vs tri-plane).** We compare hybrid representations employing tri-planes or voxel grids as discrete data structures.

| Method | Input | *UDF* | | | *SDF* | *OF* |
| | | ModelNet40 | ShapeNet10 | ScanNet10 | Manifold40 | ShapeNet10 |
| --- | --- | --- | --- | --- | --- | --- |
| CNN | Tri-plane | 82.2 | 92.1 | 63.4 | 82.5 | 88.4 |
| 3D CNN | Voxel | 83.7 | 91.6 | 68.1 | 81.1 | 85.2 |
| Transformer | Tri-plane | **87.0** | **94.1** | **69.1** | **86.8** | **91.8** |

Table 8: **Neural field classification results (voxel grid vs tri-plane).** We compare hybrid representations employing tri-planes or voxel grids as discrete data structures.

experimentation, as it strikes a suitable balance between creating reconstructions that are neither overly sparse nor excessively dense.

**Images from *RF*.** Given a camera pose and intrinsic parameters, to render images from a *RF*, we employ the same volumetric rendering scheme as in Li et al. (2023), outlined in Appendix A. An overview of the procedure is the following: for each pixel location, we cast the corresponding ray from the camera and sample 3D points along the ray. We feed these coordinates to a $(T, M)$ pair, obtaining the corresponding $(R, G, B, \sigma)$ field values. Then, we compute the volumetric rendering equation to calculate the final RGB value of the image pixel.

### B.2 EXAMPLES OF RECONSTRUCTIONS BY TRI-PLANES

In Fig. 4, we report some examples of point clouds, meshes, voxel grids and images reconstructed from tri-plane neural fields fitted to *UDF*s, *SDF*s, *OF*s and *RF*s, respectively. For all fields, we can observe a very good reconstruction quality.

### B.3 COMPARISON BETWEEN RECONSTRUCTIONS BY NEURAL FIELD PROCESSING FRAMEWORKS

In Fig. 5 and Fig. 6, we show reconstructions of point clouds and meshes obtained by different frameworks used to process neural fields. We can notice that neural fields in which the neural component is a shared network trained on the whole dataset, i.e. Functa and DeepSDF, cannot properly reconstruct the original explicit data. Conversely, methods relying on fitting an individual network, either a large MLP (inr2vec) or a tri-plane and a small MLP (ours), provide high-quality reconstructions.

## C VOXEL GRID HYBRID NEURAL FIELDS

In this paper, we use tri-plane neural fields. However, other kinds of hybrid neural fields may be considered plausible alternatives. Thus, we investigate the employment of voxel grid hybrid neural fields. First, we analyze their reconstruction quality and memory footprint. We report results on the Manifold40 test set in Table 7. We observe that, compared to tri-planes, voxel-based hybrid fields achieve comparable reconstruction accuracy but at the cost of a much larger number of parameters, i.e., 554K vs 64K.

Then, we also try to classify voxel grid hybrid neural fields. With our resources (an RTX 3090 GPU), we were not able to train the Transformer architecture used for tri-planes. Indeed, using a voxel grid would require tokens of size $32^3$ in our formulation (Transformer invariant to the channel

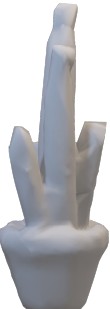
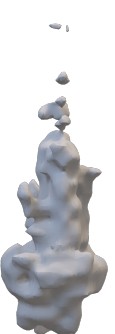

Figure 7: **Tri-plane channel shuffling. Left:** Mesh reconstructed from a *SDF* tri-plane neural field. **Right:** Mesh reconstructed by spatially shuffling the features of the tri-plane while keeping the original MLP.

order), leading to prohibitive training resources. Thus, we employ a 3D CNN (ResNet50-style). Results are reported in Table 8. We can see that although processing voxels with a 3D CNN is more expansive in terms of both storage and computation than processing tri-planes with a 2D CNN, they achieve comparable classification performance (row 1 vs 2) without a clear winner between the two approaches. Yet, thanks to their high compactness, we can process tri-planes by Transformers, achieving significantly better performance (last row). Finally, we highlight that though the performances yielded in this experiment by a hybrid voxel-based approach are inferior to tri-planes, they are still much better than those achievable by previous proposals (see Table 2), which vouches for the effectiveness of hybrid approaches when it comes to processing neural fields without compromising reconstruction accuracy.

## D    DEEPER INVESTIGATION ON THE TRI-PLANE AND MLP CONTENT

In this section, our objective is to gain a deeper understanding of the information stored in the tri-plane and the MLP.

### D.1    IS THE MLP ALONE ENOUGH FOR RECONSTRUCTION?

The first question we aim to address is whether the MLP alone can serve as a neural field capable of representing 3D shapes without relying on the features provided by the tri-plane. As explained in Section 3, we utilize the 3D coordinates of a point to retrieve the corresponding feature vector from the tri-planes. This latter is then concatenated with the positional encoding of the coordinates. To examine whether the 3D coordinates and the MLP alone are sufficient to obtain accurate outputs, we conduct an experiment where we shuffle the tri-plane features along the spatial dimensions while preserving the channel orders. This means that each point $\mathbf{p}$ will be associated with a different yet meaningful feature vector from another 3D point. If the MLP is still capable of providing the correct output in this scenario, it implies that all the geometric information of the underlying 3D shapes is likely contained within the MLP, and the tri-plane is actually not necessary. Thus, we conducted a reconstruction experiment of meshes from *SDF*, similar to that reported in Table 1. We compared the reconstructed mesh with the ground truth by calculating the Chamfer Distance and the F-score, using 16,384 points sampled from the surface. We note that when utilizing the MLP with features shuffled spatially, we get significantly worse values for both the Chamfer Distance and the F-score (2.9mm vs 0.16mm), which indicates poor reconstruction quality. This is clearly visible in Fig. 7. This outcome highlights that the MLP alone, without the tri-plane features, is not capable of accurately reconstructing the original shape. Thus, we believe that the tri-plane provides essential geometric information about the represented 3D shape.

|                                | | SDF |
| ------------------------------ | --------------- | ---------- |
| Method                         | Input           | Manifold40 |
| MLP                            | MLP (Tri-plane) | 4.3        |
| NFN (Zhou et al., 2023a)       | MLP (Tri-plane) | 4.1        |
| NFT (Zhou et al., 2023b)       | MLP (Tri-plane) | 4.1        |
| inr2vec (De Luigi et al., 2023) | MLP (Tri-plane) | 7.4        |
| Ours                           | Tri-plane       | **86.8**   |

Table 9: **Classification of MLPs of tri-plane neural fields on the Manifold40 test set.** Neural fields were randomly initialized.

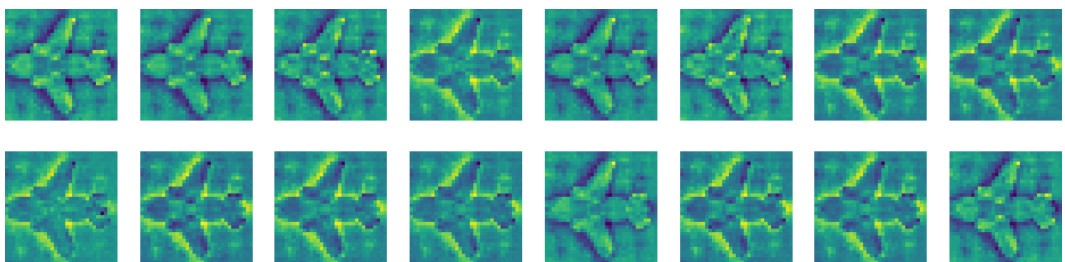

Figure 8: **Channel visualizations.** We select one of the tri-planes and visualize all its 16 channels learned for an airplane.

### D.2 Is the MLP alone enough for classification?

We conducted additional experiments where we treated each MLP associated with a tri-plane as input to a classification pipeline. We utilize various methods, including a simple MLP classifier, as well as advanced frameworks such as NFT and inr2vec, which directly process MLPs. In this case, we completely disregarded the information provided by the tri-plane. We performed this experiment starting from random initialization. As shown in Table 9, even though MLPs used in tri-planes are smaller than those used in other experiments, we note that directly classifying them leads to unsatisfying performances.

### D.3 Tri-plane channel visualizations

To understand what information is contained in different channels of a tri-plane feature map, we try to visualize all of them for a fixed shape. Fig. 8 shows all 16 channels of a *SDF* tri-plane feature map of a *SDF* neural field representing an airplane. We can see that, although values can change across channels, the overall shape is outlined and repeated in each channel. This also motivates why a CNN network that is not invariant to the channel order can work on such kind of input.

### D.4 Channel order investigation

In this section, we further analyze the tri-plane channel content to highlight that the main difference between two training episodes concerning the same shape boils down to a permutation of the channel order. Thus, we conduct a similar experiment to that illustrated in Fig. 3 with a tri-plane with only 8 channels. We visualize results in Fig. 9, also reporting the tri-plane channel visualization for both training episodes, $A$ and $B$, as well as the permutation that aligns the channels found in $A$ to $B$. We discovered that permutation by minimizing the reconstruction error. We can appreciate that after the permutation, the corresponding channels contain similar features, resulting in the possibility of reconstructing the shape with the $M_B$.

### D.5 Additional visualizations

In Fig. 10, we show additional tri-plane visualization, similar to those of Fig. 3 (left).

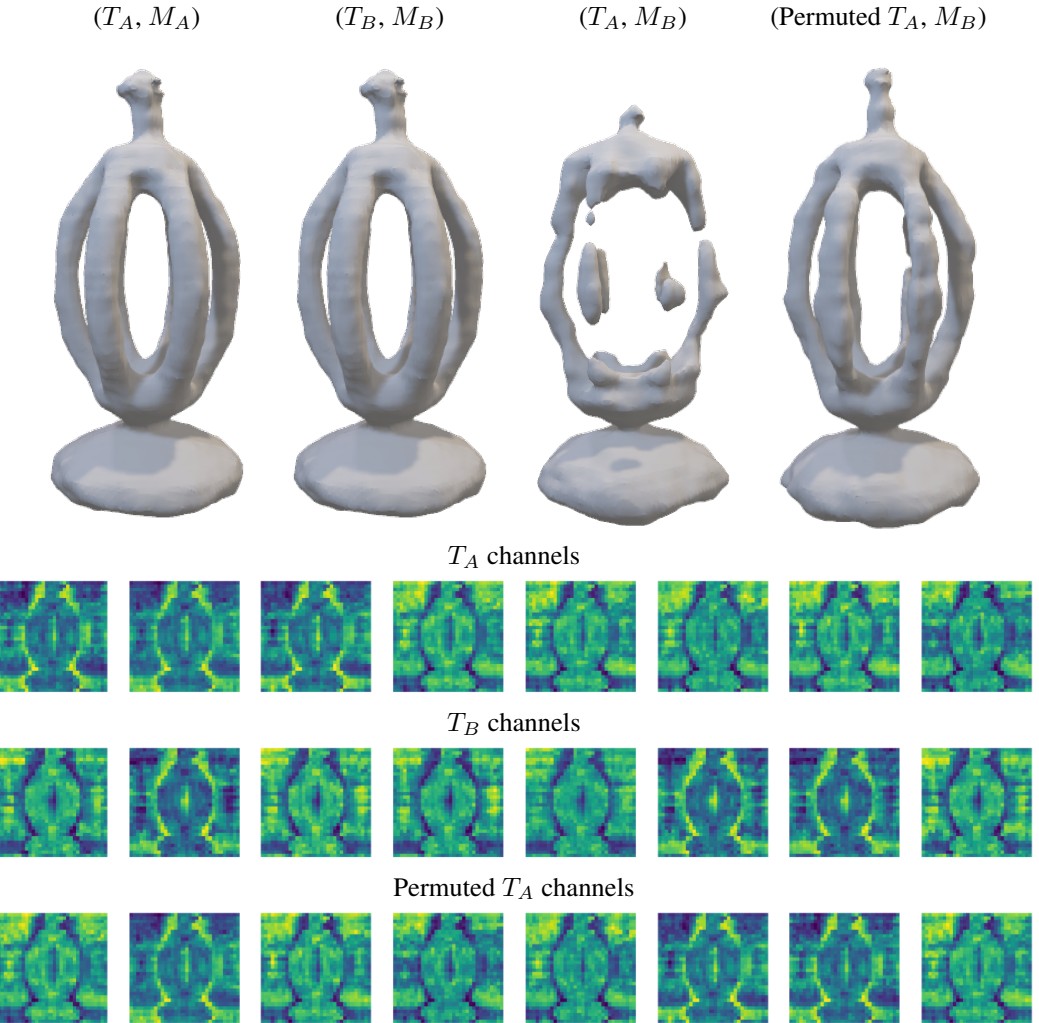

Figure 9: **Row 1:** Reconstructions of different combinations of (tri-plane, MLP) pairs with different initializations. Permuted $T_A$ means that the channels of $T_A$ are permuted in order to minimize the reconstruction error when combined with $M_B$. **Rows 2–4:** Channel visualizations.

# E    IMPLEMENTATION DETAILS

## E.1    DATASETS

Regarding single MLP neural fields based on SIREN networks, we employ the same datasets shared by De Luigi et al. (2023) with the same train/val/test splits for *SDF*, *UDF*, and *OF*. When creating the tri-plane neural field datasets, we employ the original explicit datasets used to create the inr2vec De Luigi et al. (2023) neural fields. We also follow the same offline augmentation protocol that employs a non-uniform scaling along the three axes and then a re-normalization of the shape into the unit sphere to reach roughly 100K shapes in total. Thereby, the same explicit data is used to build the benchmark, while the only aspect that varies is the way to represent the neural field, i.e., MLP only vs (tri-plane, MLP). Finally, to learn *RF*, we adopt the dataset introduced in Xu et al. (2019). This dataset contains renderings from 13 classes of ShapeNet, and for each shape, 36 views are generated around the object and used to fit the *RF*. In this case, we use the original dataset that accounts for 40511 shapes with no prior augmentation. For the train/val/test splits, we randomly sample 80% of the objects for the training set and select 10% shapes for both the validation and test splits.

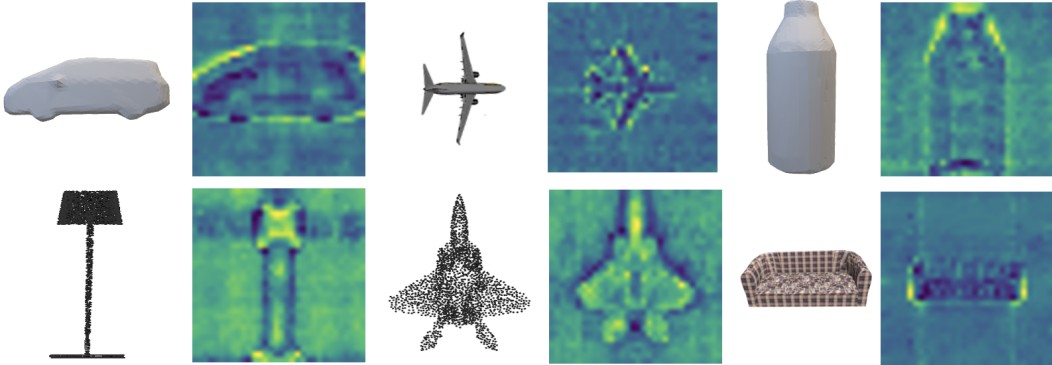

Figure 10: **Additional visualizations.** For each object, we visualize the explicit reconstruction obtained from the hybrid neural field and one tri-plane feature map for three different fields, *SDF*, *UDF*, and *RF*. We visualize features as the normalized sum of all channels with a viridis colormap.

We additionally note that the ShapeNet10 and ScanNet10 datasets mentioned in the main text are subsets of 10 (shared) classes of ShapeNet (Chang et al., 2015) and ScanNet (Dai et al., 2017), respectively, originally proposed in Qin et al. (2019).

### E.2 BENCHMARK

In this section, we detail some implementation details and choices behind the results reported in Table 2 by specifying how experiments were run for each one of our competitors.

**DeepSDF.** DeepSDF (Park et al., 2019) was originally intended as a shared network architecture that learns an *SDF* from a dataset of meshes. The results reported in Table 2 for Manifold40 (Hu et al., 2022) were therefore obtained by running the code from the official repository to train the framework for 100 epochs and compute the 1024-dimensional embeddings of training, validation, and test set. A classifier (3 fully connected layers with ReLUs) was then trained on those embeddings. For *UDF* results, instead, we trained the framework on point cloud datasets by replacing the DeepSDF loss with the binary cross entropy of Eq. (1). Extensions to *OF* and *RF*, however, are not as straightforward and were thus not included as part of our work.

**inr2vec.** The inr2vec framework (De Luigi et al., 2023) was trained on each point cloud, mesh, and voxel dataset for 100 epochs via the official code and the resulting embeddings used to train a classifier (3 fully connected layers with ReLUs). The original work by De Luigi et al. (2023) deals with *UDF*, *SDF*, and *OF*; an extension to *RF* would not be trivial thus it was not covered by our experiments.

**DWSNet.** Navon et al. (2023) test their DWSNet architecture on neural fields representing grayscale images. We modified their official neural field classification code to work with *UDF*, *SDF*, *OF*, and *RF*, and trained their classifier (DWSNet + classification head) for 100 epochs.

**NFN and NFT.** We modified the official code for NFN (Zhou et al., 2023a) and NFT (Zhou et al., 2023b) to work with *UDF*, *SDF*, *OF*, and *RF*, by using two hidden equivariant layers (16 channels each) followed by an MLP classification head. We trained on each dataset for 80 epochs, by which time the validation loss had stopped improving.

**Functa.** To perform experiments with the Functa framework (Dupont et al., 2022), we start from the original code provided by authors and adapt it to fit *UDF*, *SDF*, and *OF*. In particular, we use a latent modulation of size 512. At each iteration, we use 10k points to compute the loss. The inner loop of the meta-learning process is optimized for three iterations, and in total, we optimize for $5e^5$ steps with batch size 2. After the training phase, the weights of the shared network are frozen, and we execute the inner loop of the meta-learning protocol for three steps to obtain the latent modulation vector for each field.

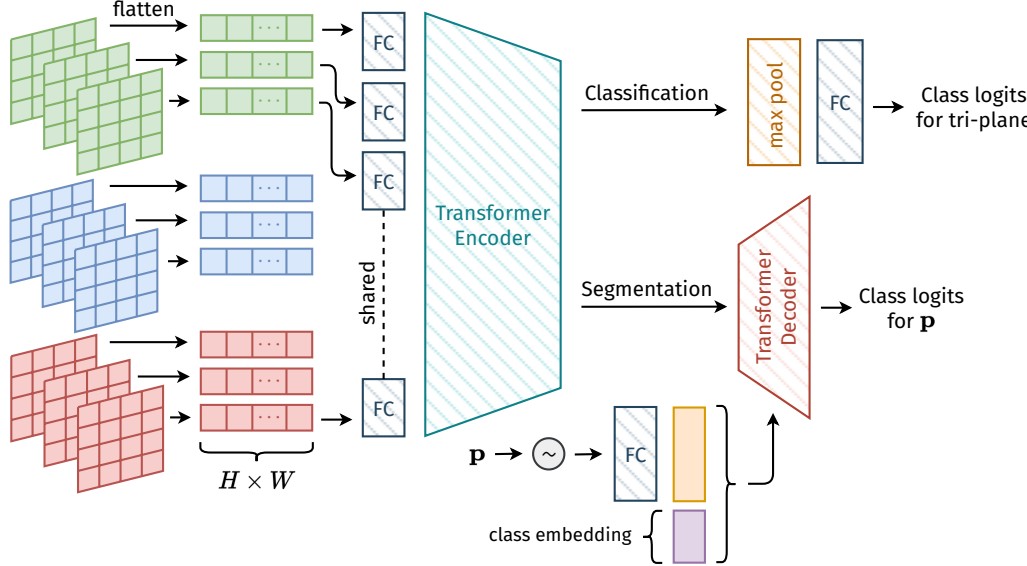

Figure 11: **Architecture for tri-plane processing**

### E.3 ARCHITECTURES

We provide here some additional details regarding the Transformer architectures used for classification and segmentation depicted in Fig. 11. In both cases, the channels of the tri-planes are flattened and linearly projected to 512-dimensional vectors and fed to the Transformer encoder, which is the original Transformer encoder proposed in Vaswani et al. (2017). In our case, the encoder consists of 4 heads with 8 layers each. The encoder produces a single embedding for each tri-plane channel, and for classification, we use a max pool operator to achieve invariance to the channel order and obtain a single global embedding. The classifier consists of a single fully connected layer that yields the predicted probability distribution for the input field. In total, the number of parameters is roughly 25M, which is very similar to the number of parameters of a Resnet50. As regards part segmentation, we attach an additional Transformer decoder composed of 2 parallel heads with 4 layers each. The decoder takes in input the sequence of tokens from the encoder and the sequence of tokens obtained starting from the coordinates query to be segmented. More precisely, each point is first encoded using positional encoding from Mildenhall et al. (2020b) that gives a 63-dimensional vector, and then we linearly project it into a 512-dimensional vector. Finally, given an object to segment, we also attach the one-hot encoding of its class (standard practice in part-segmentation). The decoder output is a sequence of transformed tokens that are given in input to a single layer to predict the final probability distribution. Note that we do not use the decoder auto-regressively as in the original Transformer. We just compute a forward pass on a batch of query points to segment all of them simultaneously. Hence, we also do not use masked attention at training time.

### E.4 TRAINING

We provide here additional training details. For classification, we use the same hyperparameters for all experiments. We adopt the AdamW optimizer (Loshchilov & Hutter, 2017), with the OneCycle scheduler (Smith & Topin, 2019) with maximum learning rate of 1e-4, and train for 150 epochs with batch size set to 256 using the cross entropy loss. We use a single NVIDIA RTX 3090 for all experiments. At training time, we apply a random crop of size $30 \times 30$ to the tri-planes that have a resolution of $32 \times 32$. As for part segmentation, we use the same configuration as for classification, although we train for 75 epochs with batch size 32. For training and testing, we use the same protocol used by our competitors. Indeed, at training time, we use the cross entropy loss function with all the object parts, although at test time, metrics are computed by selecting the highest scores among the correct object parts for a given class.

### E.5 RANDOM INITIALIZATION

Throughout the main paper, we use the term "random initialization" to differentiate our setup from that considered in De Luigi et al. (2023), where all neural fields are trained starting from the same, albeit random, set of parameters, i.e. initial values of weights and biases are sampled once and then used as the starting point to fit all MLPs. In our work, instead, we consider the more realistic scenario in which each individual neural field is trained starting from a random and different set of parameters. Nonetheless, when sampling initial values of weights and biases for each shape, we do indeed follow the specific initialization scheme suggested for SIRENs (Sitzmann et al., 2020). In other words, "random initialization" means that a different random set of parameters, sampled according to the SIREN rules, is used to initialize each MLP. Tri-planes, instead, are initialized with values sampled from a Gaussian distribution.

## F  TRAINING AND INFERENCE TIME

In this section, we provide additional details on training and inference times, computed on a single e NVIDIA RTX 3090.

| Method | *SDF* Time (hours) ↓ |
|---|---|
| DeepSDF (Park et al., 2019) | 61 |
| Functa (Dupont et al., 2022) | 120 |
| MLP | 80 |
| Tri-plane (`Ours`) | **45** |

Table 10: **Training time comparison.** Time required to fit approximately 100K shapes on the Mandifold40 training set. Times are computed on a single NVIDIA RTX 3090.

| Method | *UDF* Time (seconds) ↓ | | | |
|---|---|---|---|---|
| | 2048 points | 16K points | 32K points | 64K points |
| PointNet (Qi et al., 2017a) | **0.002** | 0.004 | 0.006 | 0.012 |
| PointNet* (Qi et al., 2017a) | 0.099 | 0.764 | 1.388 | 2.793 |
| `Ours` | 0.003 | **0.003** | **0.003** | **0.003** |

Table 11: **Inference time to classify an input shape.** * indicates that the time to reconstruct the point cloud from the neural field is included. Times are computed on a single e NVIDIA RTX 3090.

Table 10 shows a comparison between the times required to fit approximately 100K shapes on the Manifold40 (Hu et al., 2022) training set for different methods. For neural fields consisting of an MLP only and a (tri-plane, MLP) pair, denoted by "MLP" and "Tri-plane", respectively, we fit each shape for 600 steps. Functa was trained with the original hyperparameters for 500K iterations for the meta-learning outer loop. DeepSDF was trained for 100 epochs. Notice how tri-plane hybrid neural fields are the ones requiring the least amount of time to fit the dataset.

Table 11 compares the inference time required to classify an input shape with different strategies. The first row is the classification time of a PointNet processing a point cloud. In the second row, we report the inference time of PointNet, assuming we do not have an explicit point cloud available, namely PointNet*. In this case, the time includes reconstructing the point cloud from the tri-plane neural field, which would be the only data available in our scenario. Finally, the last row reports the time required by our method, i.e., a Transformer processing an input hybrid tri-plane neural field. We show times for different point cloud resolutions (2048, 16K, 32K, 64K points). We notice that the inference time of our method is constant along the resolution axis; in particular, our method is comparable to the PointNet at low resolution, whereas it becomes increasingly faster as the resolution grows. Moreover, we highlight that our method inference times w.r.t. to those of a PointNet directly

processing explicit point clouds are comparable at lower resolutions and even better at higher ones (e.g., 0.003 `Ours` vs 0.012 PointNet for 16K points).

## G  TRI-PLANE ABLATIONS

This section provides additional ablations concerning the tri-plane structure and hyperparameters. Table 12 shows that sharing the MLP across tri-planes leads to inferior results in both classification and reconstruction on Manifold40 (Hu et al., 2022). It is also worth highlighting how sharing the MLP requires its availability at test time to create new neural fields, i.e., to create new test data, limiting the deployment scenarios of our methodology, which are instead equivalent to those of discrete data structures when using a (MLP, triplane) pair for each sample. Table 13 provides a comparative analysis of classification and reconstruction results on Manifold40 (Hu et al., 2022) between tri-planes with different resolution and/or number of channels. Interestingly, the classification accuracy and reconstruction error are quite robust to the number of channels and tri-plane resolution, which therefore are not critical design hyperparameters.

| Method | Type | *SDF* Classification Accuracy (%) ↑ | Mesh from *SDF* Reconstruction CD (mm) ↓ | F-score (%) ↑ |
|---|---|---|---|---|
| Tri-plane | Shared | 84.7 | 1.57 | 42.9 |
| Tri-plane (`Ours`) | Single | **86.8** | **0.18** | **68.6** |

Table 12: **Shared vs individual MLP.** Comparison of classification and reconstruction results of tri-planes sharing all the same MLP vs when each tri-plane has its own individual MLP. Results were computed on the Manifold40 test set.

| Resolution | Channels | *SDF* Accuracy (%) ↑ | Mesh from *SDF* CD (mm) ↓ | F-score (%) ↑ |
|---|---|---|---|---|
| $32 \times 32$ | 32 | 86.3 | 0.18 | 68.6 |
| $32 \times 32$ | 16 | **86.8** | 0.18 | 68.8 |
| $32 \times 32$ | 8 | 86.4 | 0.18 | **69.2** |
| $24 \times 24$ | 16 | 86.6 | 0.18 | 68.9 |
| $40 \times 40$ | 16 | 86.4 | 0.18 | 69.0 |

Table 13: **Ablation study of tri-plane resolution and number of channels.** Second row is our choice of tri-plane size. Results were obtained on the Manifold40 test set.

| Method | Input | Accuracy (%) ↑ ModelNet40 | ShapeNet10 | ScanNet10 | Manifold40 | ShapeNet10 |
|---|---|---|---|---|---|---|
| `Ours` | Tri-plane | 87.0 | 94.1 | 69.1 | 86.8 | 91.8 |
| PointNet (Qi et al., 2017a) | Point Cloud | **88.8** | 94.3 | 72.7 | – | – |
| PointNet* (Qi et al., 2017a) | Point Cloud | **88.8** | **94.7** | **72.8** | – | – |
| MeshWalker (Lahav & Tal, 2020) | Mesh | – | – | – | 90.0 | – |
| MeshWalker* (Lahav & Tal, 2020) | Mesh | – | – | – | **90.6** | – |
| Conv3DNet (Maturana & Scherer, 2015) | Voxel | – | – | – | – | 92.1 |
| Conv3DNet* (Maturana & Scherer, 2015) | Voxel | – | – | – | – | **92.5** |

Table 14: **Explicit methods on reconstructed data vs original discrete data.** Classification accuracy on the test set is reported. **Top:** our method, that processes (tri-plane) neural fields. **Bottom:** Methods that process explicit discrete 3D data, either reconstructed from the corresponding neural field (without *) or by operating directly on the original data (with *).

## H    EVALUATING ON THE ORIGINAL DISCRETE 3D REPRESENTATIONS

In Table 3 of the main paper, we evaluate the competitors on the data reconstructed from the tri-plane neural fields fitted on the test sets of the considered datasets since these would be the only data available at test time in the scenario described in Section 1, originally proposed by De Luigi et al. (2023), where neural fields are used to store and communicate 3D data. Nonetheless, we compare the classification performance of methods operating on discrete data both when the original data is used and when it is, instead, reconstructed from the corresponding neural field. Table 3 shows slightly better results when the original data is used. However, the difference is small enough to prove that no significant loss of information occurs when storing data with tri-plane neural fields.

## I    STUDY ON THE MEMORY OCCUPATION OF NEURAL FIELDS

| Dataset | Representation | Num Shapes | Memory (GB) |
|---|---|---|---|
| ModelNet40 | Point Cloud | 110054 | 2.52 |
| ModelNet40 | Siren MLP (UDF) | 110054 | 327.99 |
| ModelNet40 | Tri-plane (UDF) | 110054 | 20.15 |
| ShapeNet10 | Point Cloud | 103230 | 2.36 |
| ShapeNet10 | Siren MLP (UDF) | 103230 | 307.65 |
| ShapeNet10 | Tri-plane (UDF) | 103230 | 18.90 |
| ScanNet10 | Point Cloud | 108360 | 2.48 |
| ScanNet10 | Siren MLP (UDF) | 108360 | 322.94 |
| ScanNet10 | Tri-plane (UDF) | 108360 | 19.84 |
| Manifold40 | Mesh | 100864 | 10.79 |
| Manifold40 | Siren MLP (SDF) | 100864 | 300.60 |
| Manifold40 | Tri-plane (SDF) | 100864 | 18.47 |
| ShapeNet10 | Voxel | 103230 | 25.20 |
| ShapeNet10 | Siren MLP (OF) | 103230 | 307.65 |
| ShapeNet10 | Tri-plane (OF) | 103230 | 18.90 |
| ShapeNetRender | Training Images | 40511 | 204.45 |
| ShapeNetRender | Siren MLP (NeRF) | 40511 | 120.73 |
| ShapeNetRender | Tri-plane (NeRF) | 40511 | 7.42 |

Table 15: **Training dataset memory occupancy**

Nowadays, datasets are typically made of data represented explicitly, e.g., a point cloud dataset, a multi-view image dataset, and so on. However, in our vision, these datasets might be replaced by their neural field counterparts, i.e., each element would be represented as a neural field. Thus, this section investigates the trade-off of using these novel representations regarding memory occupancy.

First, we investigate the memory in GB required by each raw dataset employed in our paper. We report the results in Table 15 either by their original explicit representation or with neural fields. As neural field representation, we show the SIREN MLP adopted by our competitors (e.g., De Luigi et al. (2023)) or our tri-plane representation. We note that the tri-plane representation requires more memory than point clouds and meshes (e.g., ModelNet40 and Manifold40). Yet, the advantages of using tri-plane representations stand out when compared with either voxels (ShapeNet10) or images (ShapeNetRender). Notably, tri-planes take significantly less memory than Siren MLPs.

However, we argue that the real advantages of using neural fields are related to the memory occupied by the data being decoupled from its spatial resolution. Thus, in Fig. 12, we study the number of parameters required for different explicit representations compared to those of neural fields by varying the spatial resolution of the data. We include all variables required by an explicit representation in their parameter number, e.g., each point of a point cloud has 3 parameters: its 3 coordinates $x$, $y$, and $z$. The blue, red, and green lines represent the parameters of the explicit, the SIREN MLP, and the tri-plane representation when changing the resolution. Regarding meshes and point clouds, we

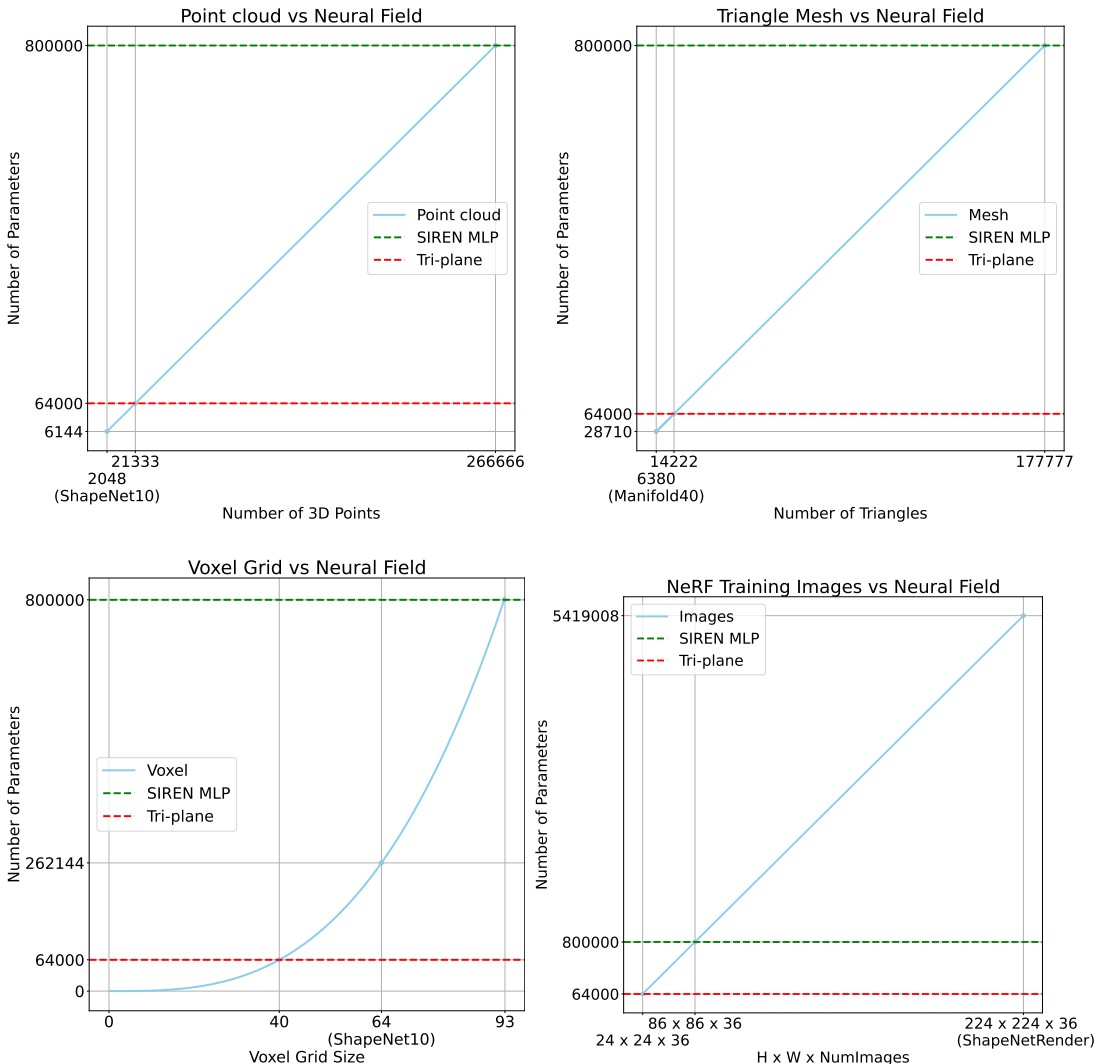

Figure 12: **Number of parameters in relation to spatial resolution: neural fields vs explicit representations**

notice that the space occupied by tri-plane neural fields is slightly larger for the resolutions used in the datasets. However, even with a point cloud of 21,333 points and a mesh with 14,222 faces, using tri-plane neural fields to represent the data becomes advantageous. This is of utmost importance, considering that real datasets could contain point clouds or meshes with many more points or faces e.g., Objaverse (Deitke et al., 2023) features meshes with more than $10^7$ polygons. The advantages are even more significant in the case of voxel grids, in which the memory occupancy scales cubically with the spatial resolution or NeRFs in which many training images per sample are required. Finally, we point out that the representation with tri-plane neural fields is advantageous compared to that with SIREN MLPs.

