# OpenReview forum: "Neural Processing of Tri-Plane Hybrid Neural Fields"
_ICLR.cc/2024/Conference — ICLR 2024 poster_

### Official Review · Reviewer_hFtS · 2023-10-23

**Soundness:** 3 good
**Presentation:** 2 fair
**Contribution:** 3 good
**Rating:** 3
**Confidence:** 5

**Summary:**

This paper defines a benchmark that covers a set of fields including occupancy, signed and unsigned, and radiance fields. The authors show that applying well established archtieures on triplanes achieves better results that processing neural fields realized as a large MLP.

**Strengths:**

1. A benchmark for triplane neural field classification.

2. The motivation of creating this benchmark is interesting and makes sense.

**Weaknesses:**

1. The paper is difficult to follow.

2. The presentation has room to improve.

3. The proposed method performs worse than existing point cloud methods as shown in Table 5.

4. It is not obvious on the advantage of the proposed method over methods working with point clouds and / or meshes.

5. How well does the proposed method generalize to unseen scenes?

6. Does the proposed method have a faster run time compared to mesh / point cloud methods?

**Questions:**

See the questions above.

---

> ### Author Response · Authors · 2023-11-17
> **Response to reviewer hFtS**
>
> We thank the reviewer for the feedback. We provide comments on his/her concerns below.
>
> __W1. The paper is difficult to follow.__ AND __W2. The presentation has room to improve.__
>
> Thank you for pointing these weaknesses out. We will be glad to submit an improved version of the paper if you could kindly provide us with more actionable feedback, e.g. pointing us to the unclear paragraphs or describing the parts of the paper that suffer from poor presentation.
>
> __W3. The proposed method performs worse than existing point cloud methods as shown in Table 5.__ AND __W4. It is not obvious on the advantage of the proposed method over methods working with point clouds and/or meshes.__
>
> Thank you for pointing out a possible source of confusion. The aim of our paper is not to set a new state-of-the-art in point cloud/mesh processing. As done in a few other very recent papers (Dupont et al. ICML22, De Luigi et al. ICLR23, Zhou et al., NeurIPS23a/b, Navon et al. ICML23), we provide an answer to a different research question, i.e. whether and how it is possible to leverage neural architectures to process data encoded as neural fields. Neural fields are functions defined at all spatial coordinates, parameterized by a neural network, which have seen increased adoption in recent years to model a variety of data (images, shapes, radiance fields, etc.) due to their intrinsic continuity and the decoupling of memory requirements from resolution. Hence, the reason to provide an answer to the former question is twofold: (i) it is a novel and challenging research problem that requires to process neural networks with neural networks; (ii) as data are increasingly stored as neural fields, the cited papers conjecture that datasets consisting only of neural fields, i.e. shipped *without* the accompanying discrete data used to create them, will become available, and processing them with deep learning models will be of practical interest. For instance, this is already the case for NeRFs, which are stored and communicated without the original training images and, indeed, do exist only as neural fields.  Our work sets a new state-of-the-art in processing neural fields, outperforming all works addressing the same scenario, as shown in Table 2 and the first two rows of table 5. The aim of Table 3 and the last three rows of Table 5 is to show that by processing a variety of neural fields with the same network architecture we are able to achieve for the first time performance comparable to specialized architectures engineered to process discrete signals like point clouds, meshes, etc.
>
> __W5. How well does the proposed method generalize to unseen scenes?__
>
> Results reported in Table 2 are on an unseen test set of neural fields, which is part of our benchmark and will be publicly released upon acceptance. Our method provides the best generalization.
>
> __W6. Does the proposed method have a faster run time compared to mesh/point cloud methods?__
>
> Yes, as shown by the inference times reported in the newly added Table 11 in Appendix F. Indeed, in our scenario data are available only as neural fields and thus running methods designed to process meshes or point clouds requires to reconstruct the discrete signal from the field, a step that makes classification/segmentation slower. It is worth noting that our method not only is faster, but it also enjoys constant run-time regardless of the point cloud resolution. Thanks to this decoupling, if one considers a different scenario where the discrete data structure is readily available, we are comparable to PointNet at low resolution, and again faster when resolution is increased.

---

### Official Review · Reviewer_fdy7 · 2023-10-30

**Soundness:** 4 excellent
**Presentation:** 3 good
**Contribution:** 3 good
**Rating:** 8
**Confidence:** 4

**Summary:**

The authors propose to use optimized triplanes for downstream tasks on neural fields, such as 3D object classification or part segmentation on various neural field types, such as unsigned distance fields, signed distance fields, occupancy fields, or radiance fields. It is shown that using triplanes in this scenario leads to a clearly better trade-off between reconstruction quality and downstream task accuracy, in comparison to previous works that utilize a latent code from a shared MLP or MLP weights as descriptors.

Further, the authors expose that architectures that are invariant to channel order in the fitted representations are important to achieve optimal accuracy in downstream tasks.

Also, the authors provide a benchmark for downstream tasks on triplane representations, which they plan to make publicy available.

**Strengths:**

- The idea of using fitted triplanes as embeddings for downstream tasks is simple but seems effective and has not been analyzed before to my knowledge.
- The results clearly show the better trade-off compared to previous methods.
- The method is evaluated on a diverse set of tasks and function representations.
- The insight regarding channel invariance is interesting and leads to the conclusion that transformers are better than CNNs, which seem unintuitive at first.
- The authors provide a benchmark datasets for the community to test architectures on.
- The paper is well written and easy to understand.

**Weaknesses:**

- The idea of using fitted triplanes for downstream tasks like classification is "obvious" in a sense.
- There is the general question of what the relevant application of the proposed approach might be. This is a problem for all methods that aim to solve downstream tasks on optimized neural field representations. Usually, data (images or point clouds, etc) was used to obtain the neural field in the first place. Solving the downstream tasks on this input representation instead of the neural field usually leads to better results, which is also confirmed by an experiment in the paper. The gap is reduced a lot though, in comparison to previous works.
- I think the term "universal neural field classifier" (which the authors claim their method to be) is misleading, as the method is not for all neural fields but only for those represented as triplanes.

**Questions:**

The use of instance-specific MLPs as decoders for triplanes makes sense in terms of reconstruction quality. However, this also leads to some information being represented in the MLP, instead of the triplane. I wonder how the downstream task quality is changing when a shared MLP is used. This experiment seems to be missing in the paper and I would be interested to see the comparison.

---------
I thank the authors for the replies to my concerns and also for elaborating on the general motivation of the research direction.
The main critique of other reviewers seem to go into a similar direction - questioning the usefulness of the proposed approach. I still agree to some degree but I also see that there might be potential applications in the future. That aside, I think this paper does something novel and analyses it well, which is why I will keep my score.

---

> ### Author Response · Authors · 2023-11-17
> **Response to reviewer fdy7**
>
> We thank the reviewer for the very positive judgment of our work and address his/her concerns below:
>
> __W1. The idea of using fitted triplanes for downstream tasks like classification is "obvious" in a sense.__
>
> We agree that the idea of performing tasks on hybrid neural fields by processing their discrete component only is, to some extent, a natural development of the increasing research interest in these new representations. However, we believe that the notion that a tri-plane encodes enough geometric information to classify and segment the underlying signal cannot be taken for granted.  To the best of our knowledge, we are the first to experimentally validate this claim, which has proven to be conducive to a straightforward and effective method for the challenging and novel problem of neural processing of neural fields.
>
> __W2. There is the general question of what the relevant application of the proposed approach might be. This is a problem for all methods that aim to solve downstream tasks on optimized neural field representations. Usually, data (images or point clouds, etc) was used to obtain the neural field in the first place. Solving the downstream tasks on this input representation instead of the neural field usually leads to better results, which is also confirmed by an experiment in the paper. The gap is reduced a lot though, in comparison to previous works.__
>
> As described in the introduction, we embrace the vision, first proposed by [De Luigi et al. ICLR23], according to which neural fields will become a standard method to store and communicate 3D information, with repositories hosting digital twins of 3D objects and scenes stored as neural networks becoming commonly available. This will allow to decouple the memory cost of the representation and its spatial resolution (as a surface with arbitrarily fine resolution can be represented with a fixed number of parameters, i.e. the weights of the corresponding neural network), while also holding the potential to provide a unified framework to represent 3D information (instead of differentiating between point clouds, meshes, voxel grids, and scene radiances). In this setting, the original “explicit” data on which the neural field was originally trained might not be available to the user, who accesses the 3D object by downloading the weights of the corresponding neural network.   For instance, this is already the case for NeRFs, which are stored and shipped without the original training images and, indeed, do exist only as neural networks. Therefore, we believe that devising methods that solve tasks by processing neural fields as effectively as those who work on the original data is the key to unlock the benefits of the neural field representation and allow the aforementioned vision to become a reality.
>
> __W3. The term "universal neural field classifier" (which the authors claim their method to be) is misleading, as the method is not for all neural fields but only for those represented as triplanes.__
>
> Thank you for pointing this out, we agree that it can be misleading. We replaced it with “universal tri-plane classifier”.
>
> __Q1. I wonder how the downstream task quality is changing when a shared MLP is used. This experiment seems to be missing in the paper and I would be interested to see the comparison.__
>
> We added classification results in the revised version of the paper, in Table 12, Appendix G. Sharing the MLP leads to worse classification performance, which drops by more than 2 points. We also point out that a major limitation of deploying a shared MLP is the significantly worse reconstruction quality, as shown in Table 1 of the paper.  It is also worth highlighting how sharing the MLP requires its availability to create new neural fields, i.e. to create new test data, limiting the deployment scenarios of our methodology, which are instead equivalent to those of discrete data structures when using a (MLP, triplane) pair for each sample.

---

### Official Review · Reviewer_ue4V · 2023-10-31

**Soundness:** 3 good
**Presentation:** 3 good
**Contribution:** 3 good
**Rating:** 6
**Confidence:** 4

**Summary:**

The paper investigates the utilization of learned neural fields as a means of representing objects for classification and part segmentation tasks. The authors propose a hybrid NeRF (Neural Radiance Field) that combines a tri-planes data structure with an MLP (Multi-Layer Perceptron) for object encoding. This hybrid approach encompasses various fields representations, including Sign/Unsigned Distance Functions (SDF/UDF), Occupancy, and Radiance fields.

The experiments conducted in the study reveal that the representation learned through tri-plane parameters remains nearly identical (up to channel permutation) even when the NeRF is trained on the same data but with different random initializations. This robustness greatly simplifies the deployment of this method in comparison to previous approaches. Additionally, the experiments demonstrate that the classification and part segmentation performance achieved through tri-plane features is on par with specialized models designed for processing explicit object representations, such as point clouds or meshes.

**Strengths:**

- The proposed approach introduces a versatile method for encoding object representations across various neural fields.

- Classification using learned tri-plane features demonstrates superior performance compared to other existing NeRF encoding methods that rely solely on MLP parameters.

- The authors also explore different techniques for reshaping tri-plane feature tensors to ensure that predictions remain invariant to channel permutations.

**Weaknesses:**

- The proposed method necessitates per-object optimization to acquire individual tri-plane features. The author conducted a comparative analysis of object reconstruction using this technique against other solutions, which employed a shared network trained on the entire dataset, like Functa (Dupont et al.). They reported the performance and the number of parameters (see Table 1). However, it is worth noting that the required computational resources, particularly in terms of training time, have not been explicitly reported or discussed.

- In the ablation study, as presented in Table 6, the primary focus is on investigating various architectural aspects related to the classification of the learned tri-plane representations. While this provides valuable insights, it's important to highlight that certain variations within the proposed method, such as utilizing a shared MLP with distinct tri-planes across data, adjusting spatial resolution, or varying the number of channels within tri-plane structures, have not been subjected to ablation analysis. Addressing these aspects could provide a more comprehensive understanding of the method's performance and potential optimizations.

**Questions:**

- How do you explain the relatively lower performance of model when is trained and tested on the same neural field, like UDF and SDF as shown in Table 4? This seems to be counterintuitive.

- As experiments show (e.g. Fig.3 right), the object representation mostly is encoded by the tri-plane parameters, rather than the MLP. This raises a question of whether the MLP can be shared across data samples for efficiency?

---

> ### Author Response · Authors · 2023-11-17
> **Response to reviewer ue4V**
>
> __W1. “The proposed method necessitates per-object optimization to acquire individual tri-plane features. The author conducted a comparative analysis of object reconstruction using this technique against other solutions, which employed a shared network trained on the entire dataset, like Functa (Dupont et al.). They reported the performance and the number of parameters (see Table 1). However, it is worth noting that the required computational resources, particularly in terms of training time, have not been explicitly reported or discussed.”__
>
> Thank you for highlighting a missing dimension in our comparison. We ran additional experiments to measure the difference in fitting time across different ways to implement neural fields. Results are reported in the revised version of the paper in Table 10, Appendix F. Triplanes are the fastest to fit an entire dataset and require roughly half the time of SIRENs and one third the time of Functa.
>
> __W2. “In the ablation study, as presented in Table 6, the primary focus is on investigating various architectural aspects related to the classification of the learned tri-plane representations. While this provides valuable insights, it's important to highlight that certain variations within the proposed method, such as utilizing a shared MLP with distinct tri-planes across data, adjusting spatial resolution, or varying the number of channels within tri-plane structures, have not been subjected to ablation analysis. Addressing these aspects could provide a more comprehensive understanding of the method's performance and potential optimizations.”__
>
> Thank you for suggesting an interesting ablation study we haven’t performed. We tested down-stream classification performance for triplanes with a shared MLP for Manifold40 (SDF), reported in the new Table 12, in Appendix G. Results show that sharing the MLP negatively affects the classification performance, which drops by more than 2 points. This result, combined with the significantly worse reconstruction quality of the shared triplane (chamfer distance  0.18 mm of single versus 1.57 mm for shared MLP configurations) suggests that an MLP for each triplane should be the preferred setup with triplanes, and is indeed the one we have used in all our experiments.
> As for varying the number of channels and the resolution of the triplane, we conducted additional ablation studies following the reviewer's suggestion and reported them in the new Table 13 in Appendix G. Interestingly, the classification accuracy and reconstruction error are quite robust to the choices of the number of channels or the resolution of the triplanes, which therefore are not critical design hyperparameters.
>
> __Q1. “How do you explain the relatively lower performance of model when is trained and tested on the same neural field, like UDF and SDF as shown in Table 4? This seems to be counterintuitive.”__
>
> Thank you very much for being such an attentive reviewer and catching this error. The strange numbers are actually due to a swap of the values between the UDF and SDF columns, which is now fixed. Indeed, due to the swap the old wrong table was not coherent with the result of our method when trained and tested on SDFs (86.8), which was also reported in table 2.
>
> __Q2. “As experiments show (e.g. Fig.3 right), the object representation mostly is encoded by the tri-plane parameters, rather than the MLP. This raises a question of whether the MLP can be shared across data samples for efficiency?”__
>
> As discussed above and validated by the results in Table 1 and the additional experiment reported in Table 12 of Appendix G, sharing the MLP significantly lowers the reconstruction quality and also negatively affects the downstream performance. The saving in terms of space are limited, as the MLP only has 15k parameters, while the time efficiency is exactly the same. Moreover, sharing the MLP requires its availability to create new neural fields, i.e. to create new test data, limiting the deployment scenarios of our methodology, which are instead equivalent to those of discrete data structures when using a (MLP, triplane) pair for each sample. Hence, we believe the advantages of sample-specific MLPs to outweigh the limited disadvantages and it is the configuration we recommend.

---

### Official Review · Reviewer_3J4Y · 2023-11-01

**Soundness:** 3 good
**Presentation:** 3 good
**Contribution:** 4 excellent
**Rating:** 5
**Confidence:** 4

**Summary:**

The paper presents a method to encode discrete 3D information into the effective continuous triplane, allowing for the larger vision transformer to perceive in 3D. The paper validates their methods in both classification and segmentation tasks -- much better then previous version that used MLP.

**Strengths:**

I like the topics this paper explores. Instead of directly consuming raw and messy 3D data, we could represent that data using neural representation, which will make the network design much easier.

The presented method achieves the significantly improved results compared with baselines. Although the proposed method is not novel -- simply replacing MLP with the more effective triplane, it shows better performance than the network that takes in raw discrete 3D dataset, which suggests a new paradigm to deal with 3D data.

**Weaknesses:**

Parameter and time efficiency comparison is missing. We know that triplanes work better than global MLP. However, there was no free lunch. Triplane-based is usually parameter-intensive. So I’m concerned that the triplane based representation would consume lots of space compared with the original dataset. And the paper doesn’t report any comparison.

Also I notice that the paper uses the explicit extracted from the learned triplane in the classification tasks of Table3. I’m not very sure if it makes sense. Although the triplane is very effective, there's still information loss compared with original data. I would suggest the author justify it a bit.

The random initialization of triplane and MLP  is concerning. The reason is that the method uses the Sine/Cosine as the activation function whereas random initialization is not preferred as stated in other works. Instead, a specific way of random initialization was suggested in other papers like the SIREN paper. Also, the channel order with respect to the initialized value is not very clear.

Regarding the dataset, I’m not very sure what's ScanNet10 used in the paper. It seems unexplained.

In short, I tend to hold this paper and vote for weakly reject. I'm really looking forward to hearing back from authors during the rebuttal and clarify about my concerns.

**Questions:**

Please address the question above

---

> ### Author Response · Authors · 2023-11-17
> **Response to reviewer 3J4Y**
>
> We thank the reviewer for the positive judgment on the topic explored by the paper and address her/his concerns below.
>
> __W1. Parameter and time efficiency comparison is missing. We know that triplanes work better than global MLP. [...But] triplane-based is usually parameter-intensive [...and...] would consume lots of space compared with the original dataset. And the paper doesn’t report any comparison.__
>
> We provide here a more thorough comparison both in terms of parameters/memory and time.
>
> -Memory/parameters
>
> In Appendix I of the revised paper, we report a detailed comparison between the memory occupancy of triplanes, MLPs and original datasets for all the representations considered in our study. The charts in Figure 12 show that for all representations, tri-planes are much more memory efficient than MLPs. Compared to the original representations, the trade-off depends on the kind of explicit data and the resolution:  triplanes always require much less memory in the case of  NeRFs images; they compare favorably to voxels even at relatively coarse resolutions (from 40x40x40 on); in case of point clouds and meshes, triplanes start to provide memory savings from higher resolutions (about 21k for point clouds and 14k triangles for meshes).  These general considerations are confirmed if we look at the total memory used by our datasets, reported in Table 15.
>
> For instance, the original ShapeNetRenderer contains 1,458,396 images and takes 204.45GB, while the tri-plane dataset only 7.42GB. Conversely the original ModelNet40 contains 110,054 shapes and takes 2.52GB, while the tri-plane dataset requires 20.15GB. Nonetheless, we point out that the memory sizes of datasets of triplanes are perfectly affordable compared to many popular ones widely used in research, such as, for instance ImagNet1K, which includes 1,281,167 images and takes more than 100GB.
>
> -Time
>
> We include an additional analysis in terms of time required to obtain the whole dataset in Appendix F. We can see that tri-planes are the fastest to fit an entire dataset and require roughly half the time of SIRENs and one-third the time of Functa.
> We also compare the time required to fit the SDF function from a single mesh for both INRs and tri-planes. We use the same INRs used in inr2vec (MLPs with 4 layers of 512 neurons each), and the same tri-planes for all our experiments (resolution of 32x32 with 16 channels). In both cases, we optimize for 600 iterations, and at each step, we fit 50k points. The average time to fit a mesh using the tri-plane representation is 2.2 seconds on a Nvidia 3090, while doing so using an MLP takes on average 12 seconds.
> Thus, tri-planes are a better neural representation than MLPs, since they exhibit comparable reconstruction accuracy, much higher performance on downstream tasks and are more efficient in terms of both memory and time.
>
> __W2. Also I notice that the paper uses the explicit extracted from the learned triplane in the classification tasks of Table3. I’m not very sure if it makes sense. Although the triplane is very effective, there's still information loss compared with original data. I would suggest the author justify it a bit.__
>
> Thank you for raising this unclear point. To compare against 3D discrete representations, we followed the protocol established in inr2vec (De Luigi et al., ICLR23). The reason why it uses the neural field to reconstruct the discrete data is that their and our method are conceived to operate in a setting where datasets storing 3D information are themselves made of neural fields, hence no explicit data is readily available. This scenario is described in the paragraph “Neural processing of neural fields” in the introduction. Anyway, the difference between neural processing of the reconstructed shape versus the original one is minimal, as shown by the results obtained classifying directly the original shape reported in Table 14, Appendix H, which validates that the information loss of the triplane with respect to the original data is marginal.

---

> > ### Author Response · Authors · 2023-11-17
> > **Response to reviewer 3J4Y (Part 2)**
> >
> > __W3. The random initialization of triplane and MLP is concerning. The reason is that the method uses the Sine/Cosine as the activation function whereas random initialization is not preferred as stated in other works. Instead, a specific way of random initialization was suggested in other papers like the SIREN paper.__
> >
> > Thank you for pointing this out, we agree that this can be a source of confusion. We use the term “random initialization” to differentiate our setup from that considered in [De Luigi et al. ICLR23], where all neural fields are trained starting from the same, albeit random, set of parameters, i.e. initial values of weights and biases are sampled once and then used as the starting point to fit all MLPs. In our work, instead, we consider the more realistic scenario in which each individual neural field is trained starting from a random and different set of parameters. Nonetheless, when sampling initial values of weights and biases for each shape, we do indeed follow the specific initialization scheme suggested for SIRENs. In other words, “random initialization” means that a different random set of parameters, sampled according to the SIREN rules, is used to initialize each MLP. We have clarified this point in the revised version of the paper (see “Random initialization” paragraph in Section 3.2 and Appendix E.5).
> >
> > __W4. Also, the channel order with respect to the initialized value is not very clear.__
> >
> > When fitting the same shape starting from two random initializations of tri-planes/mlp pairs, the features learned in the triplanes are different. Yet, we have analyzed the actual content of the triplanes and found out that the main difference boils down to a permutation of the channel order as explained in Sec. 3.2 “tri-plane content” of the main paper. To further investigate this, and to clarify the reviewer’s doubt, we have included an additional experiment described in Appendix  D.4 (see Figure 9).
> > It is important to note that this is the key finding that has motivated us to design an architecture invariant to the channels order to process tri-plane neural fields, so that a model robust with respect to the variability introduced by random initialization can be obtained.
> >
> > __W5. Regarding the dataset, I’m not very sure what's ScanNet10 used in the paper. It seems unexplained.__
> >
> > Thank you for pointing out this missing reference. ScanNet10 is a subset of ScanNet with only ten classes, introduced in PointDAN (Qin et al., NeurIPS 2019). We have fixed the corresponding reference in the main text and clarified this point in Appendix E.1.

---

> > > ### Comment · Reviewer_3J4Y · 2023-11-22
> > > **Thanks for the feedback**
> > >
> > > Hi, thanks for the feedback and sorry for the late!
> > >
> > > I think author's feedback addressed most of my concerns. I'll adjust my rating combining with other reviewers' comments.

---

### Author Response · Authors · 2023-11-17
**Revision uploaded and responses to all reviewers submitted.**

We thank all reviewers for their time reviewing our paper and useful feedback.

We notify you that we have uploaded the revised version of our manuscript as a PDF file and answered all reviewers' concerns and questions.

---

### Meta-Review · Area_Chair_x91K · 2023-12-26

**Metareview:**

The paper proposes a method to directly perform discriminative tasks on a radiance field representation (rather having to decode to 3D), such as object classification and part segmentation. In particular, the paper shows that the tri-plane is an effective representation, achieving a better tradeoff between reconstruction quality and task accuracy, in comparison to MLP-based representation. The paper explores invariance to channel permutations, achieving robustness to different initializations of radiance field. The reviewers appreciated the central idea and contribution of the paper, along with the contribution regarding how to deal with channel permutations, and were convinced that the triplane representation provides higher performance for downstream tasks, as claimed by the paper.

Reviewer 3J4Y had suggested parameter/time efficiency and asked for clarifications regarding some experiments. The author's rebuttal was satisfactory for the reviewer (the reviewer indicated they would change score from marginal below, but did not officially do so). Reviewer ue4V expressed concerns regarding the ablation and fitting time and found an error in a table, acknowledged by the authors. Overall, reviewers ue4V and fdy7 recommend acceptance of the paper as well. In total, 3/4 reviewers recommend acceptance and the AC agrees the paper is above the bar for ICLR.

**Justification For Why Not Higher Score:**

The paper is a solid contribution. There are some open questions expressed by the reviewers and the scope of the problem may not be of broad interest.

**Justification For Why Not Lower Score:**

In total, 3/4 reviewers recommend acceptance and the AC agrees the paper is above the bar for ICLR.

---

### Decision · Program_Chairs · 2024-01-16

Accept (poster)